



# Why Did Ozone Concentrations Increase During Shanghai's Static
# Management? A Statistical and Radical Chemistry Perspective
Jian Zhu[1], Shanshan Wang[1,2], Chuanqi Gu[1], Zhiwen Jiang[1], Sanbao Zhang[1], Ruibin Xue[1], Yuhao Yan[1], Bin
Zhou[1,2,3]
[1]Shanghai Key Laboratory of Atmospheric Particle Pollution and Prevention (LAP[3]), Department of Environmental Science and
Engineering, Fudan University, Shanghai, 200433, China.
[2]Institute of Eco-Chongming (IEC), Shanghai, 202162, China.
[3]Institute of Atmospheric Sciences, Fudan University, Shanghai, 200433, China.
*Correspondence to*: Bin Zhou (binzhou@fudan.edu.cn) and Shanshan Wang (shanshanwang@fudan.edu.cn)
**Abstract**
During the period of April and May 2022, Shanghai implemented city-wide static management measures to control the spread of
the Omicron variant. Compared to the lockdown in early 2020, the static management in 2022 occurred during the high-ozone
season and lasted for a longer duration. It can be considered as a "large-scale field experiment" to study the response of ambient
ozone levels to emission reductions. During this period, we conducted comprehensive observations at Fudan University
Jiangwan Campus in the northeast corner of Shanghai. Similar experiments were also conducted during the same period in 2020
and 2021. Despite the significant reduction of approximately 30% in VOCs and around 50% in $NO_2$ due to static management in
2022, the average ozone level increased by nearly 23%. This suggests that the reduction in ozone precursors and other pollutants
did not lead to a corresponding decrease in ozone concentrations as expected. Cluster analysis of diurnal patterns of ozone
concentration revealed four distinct types of diurnal ozone variations. Cluster 3 and Cluster 4, with high ozone levels,
experienced significant increases in their share during static management, ultimately leading to an overall increase in average
ozone levels in 2022. According to the Observation-Based Model (OBM) simulation analysis, the average peak concentrations of
OH, $HO_2$, and $RO_2$ in 2022 were estimated to be $5.3 \times 10^6$, $4.9 \times 10^8$, and $2.6 \times 10^8$ molecules $cm^{-3}$, respectively, representing an
increase of over 30% compared to the levels in 2020 and 2021. Although HONO photolysis was the main contributor to the
primary source of ROx radicals, the radical cycling process remained dominant for the overall production of ROx radicals. Due
to a significant decrease in $NO_2$ concentration relative to VOCs, the average VOCs/$NO_2$ ratio increased from 1.6 in 2020 to 3.0
in 2022, which is also reflected in the radical cycling. The ratio of OH radical propagation (OH+VOCs) to termination (OH+$NO_2$)
was 2.10, higher than 1.03 in 2020 and 1.60 in 2021, indicating that the different reduction proportions of precursors led to a
higher VOCs/$NO_2$ ratio, strengthening the radical cycling. The differential reduction in precursor VOCs and $NO_2$ levels due to
static management is the underlying cause for the increase in ozone concentration in Shanghai.
**1 Introduction**
To curb the spread of Omicron variant in Shanghai, China, the local government decided to implement city-wide static
management in early April 2022. The strict two-month lockdown severely impacted the economic activities and human life of
this mega city. According to official statistics, in April and May 2022, Shanghai experienced a year-on-year decrease of 42% in
its total industrial output value. Moreover, the total volume of transported goods decreased by 30% year-on-year, with road
transport witnessing a significant drop of 64%. Additionally, the port cargo throughput decreased by 31% year-on-year. The
direct effect of such lockdown policies on air quality is a significant reduction in anthropogenic emissions, which can be



considered as an ideal experiment on emissions control in a mega-city to explore the reduction potential and response of air quality . It makes sense to take advantage of this rare yet regrettable window to study the causes and management of air pollution, especially in those countries like China that face air pollution complex.

Prior to this, China had implemented a series of nationwide lockdown measures against the occurrence and spread of the virus in early 2020. This reduction in human activity is expected to significantly reduce air pollutant emissions, as confirmed by lots of studies on lockdown in 2020 (Bao and Zhang, 2020; Huang et al., 2021; Li et al., 2021b; Liu et al., 2020; Tian et al., 2021; Wang et al., 2021; Zhang et al., 2022b). Reports on the impact of the lockdown on air quality most commonly focus on measuring nitrogen dioxide ($NO_2$) and fine particulate matter ($PM_{2.5}$) (Agarwal et al., 2020; Hua et al., 2021; Chu et al., 2021). Among these pollutants, $NO_2$ from traffic sources has shown the most significant reduction, with traffic-related $NO_2$ exhibiting the largest decrease (Rana et al., 2021). Due to the lockdown taking place during winter, which is a season of high particulate matter pollution in China, the reports on the impact of the lockdown on air quality have focused more on the changes in particulate matter. The lockdown in Shanghai in 2022 was implemented in April and May during the high-ozone ($O_3$) season and lasted for a longer duration, providing an opportunity to study atmospheric pollution primarily caused by $O_3$.

Previous extensive research has demonstrated that the formation of $O_3$ in response to its precursors is highly nonlinear, rather than linear, which presents a challenge in ozone control (Liu and Shi, 2021; Wang et al., 2017; Sillman, 1999). The COVID-19 pandemic provided a costly experiment to validate this. during the static management period in Shanghai, despite a reduction in precursor emissions, the ozone levels increased compared to the previous year. The cause of this increase is attributed to an imbalance in the reduction ratio of nitrogen oxides (NOx) and volatile organic compounds (VOCs) rather than meteorological conditions, according to satellite observation results (Tan and Wang, 2022; Xue et al., 2022). In this current study, aim is to elucidate the reasons behind the increase in ozone levels in Shanghai through a comprehensive approach involving in-situ observations, mathematical analysis, and modeling. Benefitting from our conducted routine observational campaigns, we have obtained comprehensive observational data for both the static management period and corresponding historical periods. We initiated our analysis by comparing pollutant levels and diurnal variations during the static management period with those from historical reference periods. Subsequently, by clustering diurnal ozone profile patterns, we examined the reasons behind the elevated ozone levels from a statistical perspective. Furthermore, we discuss changes in radical chemistry compared to historical periods, shedding light on the increase in ozone concentrations from a photochemical process standpoint. It is worth mentioning that, based on previous researchs and our analysis, meteorological conditions are not considered to be the primary cause of the ozone increase during the static management period, even though we acknowledge that meteorological conditions are indeed important factors influencing ozone levels. The in-depth comparison of meteorological conditions is presented in Text S1 of the Supplement.

## 2 Experimental Details and Methods

### 2.1 Location and Experimental Setup

During the static management period in April and May of 2022, we conducted comprehensive observational experiments at Fudan University's Jiangwan Campus (31.34°N, 121.51°E), located in an urban area in northeastern Shanghai, China. Similar experiments were also conducted during the corresponding period in 2020 and 2021. The ambient concentrations of $O_3$, $NO_2$, $SO_2$, HONO, and HCHO were measured using the Differential Optical Absorption Spectroscopy (DOAS) system located on the rooftop of the Environmental Science Building (Zhu et al., 2020; Guo et al., 2021; Zhu et al., 2022). Based on the given optical path length and integration time, the detection limits for $O_3$, $NO_2$, $SO_2$, HONO, and HCHO were approximately 1.3 ppbv, 0.5



ppbv, 0.1 ppbv, 0.1 ppbv, and 0.5 ppbv, respectively. The measurements were carried out with a time resolution of 5-6 minutes
and detailed fitting configurations are available in Table S1. Non-methane volatile organic compounds (NMVOCs) were
monitored in real-time using the TH-300B online monitoring instrument, which has been previously described in detail in
previous reports (Gu et al., 2022; Zhu et al., 2020). The photolysis rate of $NO_2$ ($j(NO_2)$) was measured using a filter radiometer
(Meteorologieconsult Gmbh, Germany). The meteorological parameters data such as pressure (P), temperature (TEMP), relative
humidity (RH), wind speed (WS), wind direction (WD), and boundary layer height (BLH) are derived from the European Center
for Medium-Range Weather Forecasts (ECMWF) atmospheric reanalysis product ERA5 and extracted from the nest where the
measurement site is located. The $PM_{2.5}$ data was obtained from the Yangpu environment monitoring station (31.53°N, 31.25°E)
near the measurement site.
**2.2 Observation-Based Model (OBM)**
The open-source zero-dimensional box model tool AtChem2 was used to simulate atmospheric chemical processes, which is
specifically designed for use with the Master Chemical Mechanism (MCM) (Sommariva et al., 2020). The MCM, one of most
widely used chemical mechanism for chemistry, is a near-explicit chemical mechanism which describes the degradation of
methane and 142 nonmethane VOCs and over 17000 elementary reactions of 6700 primary, secondary and radical species
(http://mcm.york.ac.uk/, last access: 16 January 2023) (Jenkin et al., 2003; Saunders et al., 2003). AtChem2 software and
documentation can be found on https://github.com/AtChem/ (last accessed on January 16, 2023).
In this study, the observed data of $O_3$, $NO_2$, $SO_2$, HONO, HCHO, NMVOCs, $j(NO_2)$, P, T, RH, and BLH were used as inputs to
constrain the model calculations. The photolysis rates of other molecules such as $O_3$, HCHO, HONO, and OVOCs can be
calculated in this model platform with the basic principle driven by the solar zenith angle and scaled by the measured $J_{NO2}$
(Sommariva et al., 2020). The removal of all unconstrained and simulated species caused by the deposition is determined by a
parameterization approach, and is determined by the accumulation of the deposition velocity of 0.01 m s$^{-1}$ within the boundary
layer (Santiago et al., 2017). The sensitivity of simulation results to the deposition velocity has been studied in previous research,
and the impact is limited (Zhu et al., 2020). The model outputs include the concentration of the hydroxyl radical (OH) and
hydroperoxy radical ($HO_2$), as well as the reaction rates at each step of the simulation process.
**2.3 Machine learning**
Two machine learning methods, k-means clustering and the stacking model, were utilized in this study. In order to investigate the
reasons for the overall increase in ozone levels during the static management period from the perspective of its diurnal variation,
the k-means clustering method was applied to cluster 24-hour time series of $O_3$ concentration (Zhang et al., 2004). Time-series
clustering is a specific application of curve clustering, which is similar to trajectory clustering in the transport of air masses
(Darby, 2005; Suris et al., 2022). The procedure for k-means clustering is as follows: (i) randomly initialize k clusters and then
calculate the cluster centroid or mean, (ii) assign each data point to the nearest cluster using an appropriate distance measure, (iii)
re-calculate the cluster centroids based on the current cluster members, (iv) repeat steps ii and iii until there is no further change.
Additionally, the stacking model was applied to address missing values in DOAS observations caused by uncontrollable factors,
ensuring the continuity and variation characteristics of the data. This step was deemed necessary for two main reasons. Firstly,
the clustering analysis of $O_3$ diurnal variation demands a continuous time series without any missing values. Secondly, compared
to the conventional method of handling missing data in the input of the OBM model through simple linear interpolation, the
stacking model preserves the diurnal variation characteristics of the data, ensuring the correct constraints on the OBM model.
The stacked model is an ensemble machine learning algorithm that consists of two levels, with two or more base models at level





0 and one meta-model at level 1. The meta-model is trained using predictions made by the base models on out-of-sample data. In
other words, data that was not used to train the base models is fed into them to make predictions. These predictions, along with
the corresponding expected outputs, form the input and output pairs of the training dataset used to fit the meta-model. The
stacking model has been previously described in detail and demonstrated good performance in Zhu et al. (2022), and the
architecture of the stacking model can also be found in Figure S1. In this study, the models for $O_3$, $NO_2$, $SO_2$, HONO, and
HCHO demonstrated good performance, as shown in Figure S2-S6 of the Supplement.
**3 Results and Discussion**
**3.1 Year-on-year changes on air quality**
Figure 1 compares the average levels of the meteorological parameters and air pollutants during the period from April to May of
2020 to 2022, while Figure 2 compares the diurnal variations. In terms of meteorological parameters, the temperature and
relative humidity, and $j(NO_2)$ during the static management period in 2022 were almost unchanged compared to the same period
in 2020 and 2021. The average temperature difference in 2022 was 6.5℃, which was similar to that of 2020 and slightly higher
than that of 2021, while the average relative humidity at noon in 2022 was also comparable to that of 2020 and was 5% lower
than that of 2021. Furthermore, we also ruled out the contribution of transport from the surrounding areas to the increase in
ozone concentration in Shanghai during the 2022 static management period (see Figure S7 and S8).The abrupt reduction of
emissions across the entire industry led to a significant decrease in primary pollutant concentrations. The average concentrations
of $PM_{2.5}$ in April and May from 2020 to 2022 were 36.8±24.1 ug m$^{-3}$, 30.0±23.1 ug m$^{-3}$, and 21.8±14.0 ug m$^{-3}$, respectively,
showing a decreasing trend over the years. And the diurnal variation profile in Figure 2d shows that $PM_{2.5}$ levels decreased
proportionally throughout the entire 24-hour period, without any particularly prominent periods of decrease. The VOCs and $NO_2$
declined by 29% and 55% respectively compared to 2020, and by 35% and 51% respectively compared to 2021. Due to the
significant decrease in $NO_2$ concentration compared to VOCs, the average ratio of VOCs/$NO_2$ has increased from 1.6 in 2020 to
3.0 in 2022. However, the precursor reduction at different magnitudes has led to an increase of approximately 23% in the
average level of ozone. The photochemical production of ozone is controlled by the non-linear chemistry of the precursors VOCs
and NOx ($NO_2$+NO). The literatures have shown that Shanghai in the spring largely operates under VOCs-limited regime (Li et
al., 2021a; Xue et al., 2022). Therefore, the reduction in VOCs during the static management period may not be enough to
counteract the titration effect of NOx, and may even alter the ozone formation regime in Shanghai. From the perspective of
diurnal variation (see Figure 2), the period with a significant difference in the magnitude of the decrease between VOCs and $NO_2$
occurred during the strong photochemical process in the morning until noon. Therefore, the VOCs/$NO_2$ ratio during the static
management period was significantly higher in the morning compared to the same period in 2020 and 2021. The weakening of
the titration of nitrogen oxides on ozone during nighttime led to significantly higher nighttime average levels during the static
management period compared to 2020 and 2021. Due to the higher $O_3$ baseline concentration and higher VOCs/$NO_2$ ratio, there
was a significant increase in overall ozone levels.




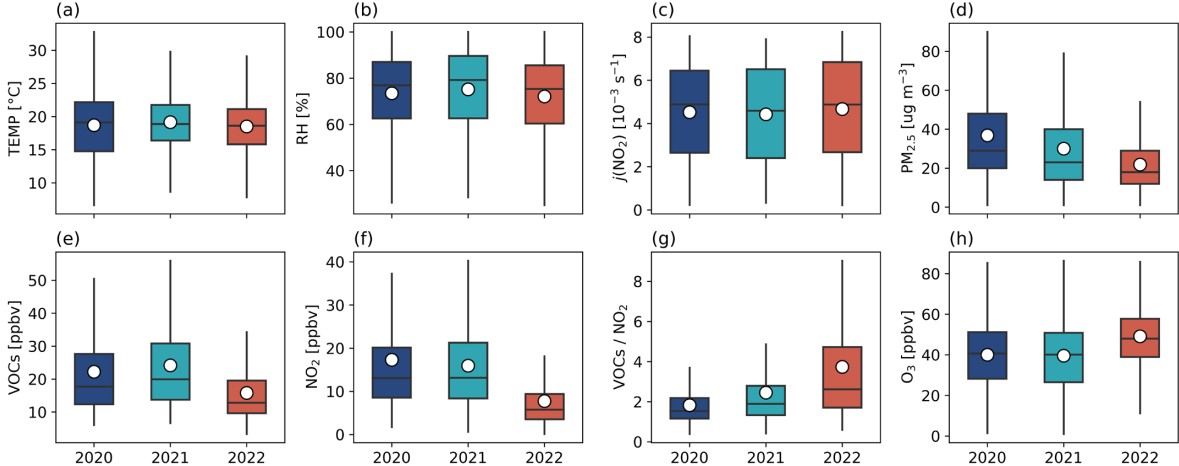

**Figure 1. Comparison of meteorological parameters (TEMP, RH, *j*(NO₂)) and air pollutants (PM₂.₅, VOCs, NO₂, VOCs/NO₂, O₃)**
**during the periods from April to May of 2020, 2021, and 2022. The top and bottom of the vertical line for each box correspond to the**
**95th and 5th percentiles, respectively. The dots represent the averages, and the top, middle, and bottom lines of the box mark the 75th,**
**50th, and 25th percentiles, respectively.**

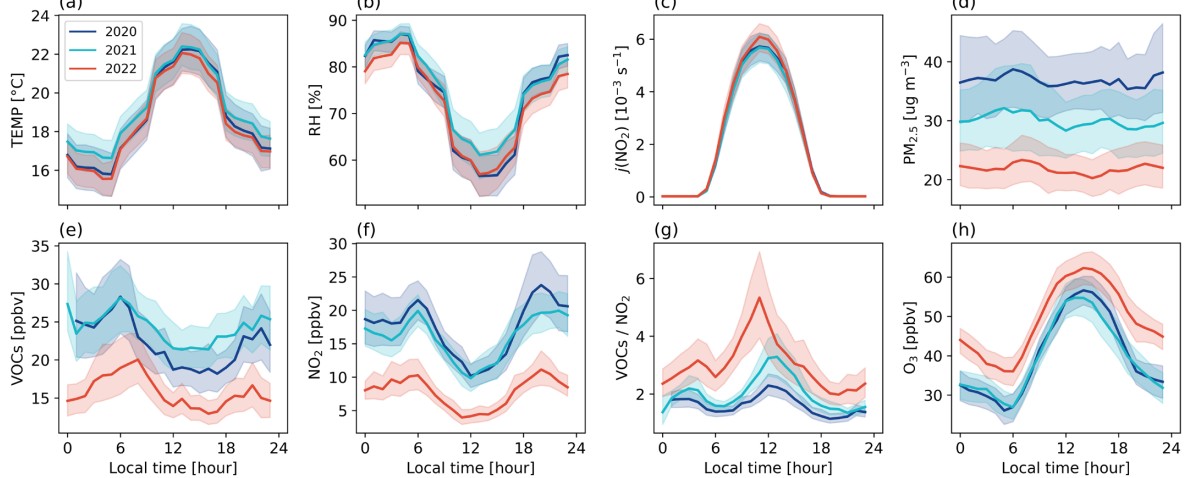

**Figure 2. The mean diurnal profiles of meteorological parameters and air pollutants during the periods from April to May of 2020,**
**2021, and 2022. Colored areas denote 95% confidence intervals.**

As VOCs are crucial precursors for ozone formation, we conducted a comparison of each VOCs component during the 2022 static management period with those of the same period in 2020 and 2021, as shown in Figure 3. We classified the 103 VOCs into six categories, including alkanes, alkenes, alkynes, aromatics, oxygenated VOCs (OVOCs), and halohydrocarbons. The detailed classification is available in the Table S2. The results revealed that aromatics experienced the most significant year-over-year reduction in absolute terms. The chemical raw materials and chemical products manufacturing industry, which is the main source of aromatics (Liu et al., 2019), accounts for 10% of the total industrial output value, and this industry experienced a 32% year-on-year decrease in total output value during the static management period. In contrast, OVOCs remained relatively stable, primarily because they are sourced from biogenic (Liu et al., 2019), and thus, were less impacted by lockdown measures relative to other VOCs.





The reduction in the imbalance of VOCs has altered the average proportion of each component. Specifically, the proportion of
aromatics decreased from 8.9% and 11.5% to 4.1%, while the proportion of OVOCs increased from 12.7% and 13.7% to 17.2%.
The photolysis of OVOCs is a major source of the important radicals ROx (OH+HO$_2$+RO$_2$) in the photochemical cycle, with a
daily average contribution rate of over 30% (Xue et al., 2016). Consequently, the rise in OVOC proportion during the static
management period has the potential to enhance the photochemical process. The mean diurnal profiles of the VOCs indicated
that the daily average concentration range in 2020, 2021, and 2022 was between 18-28 ppbv, 21-28 ppbv, and 13-20 ppbv,
respectively.
In 2022, the peak time of VOCs was observed at 08:00, which exhibited a delay compared to the peak times observed in 2020
and 2021 at 06:00, resembling the previously reported "weekend effect" on VOCs that the peak time of VOCs is delayed on
weekends in comparison to weekdays (Cai et al., 2010). This finding indicates that the reduced human activities during the 2022
period, similar to weekends, led to a decline in anthropogenic VOC emissions in the morning.

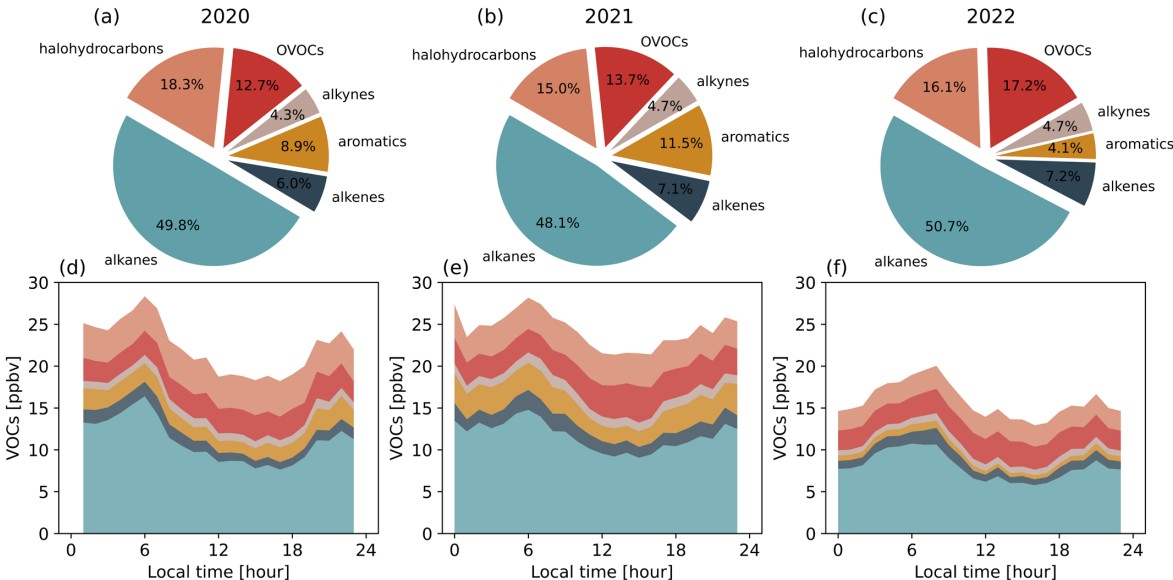


**Figure 3. The proportions (a, b, c) and the mean diurnal profiles (d, e, f) of different VOCs components during the periods from April**
**to May of 2020, 2021, and 2022.**

### 3.2 Clustering of O$_3$ diurnal profiles

The k-means algorithm clustered the ozone diurnal profiles over the three years into four types, as shown in Figure 4a. These
four types of profiles can be described as follows: Cluster 1 with low background concentration and low net production; Cluster
2 with low background concentration and high net production; Cluster 3 with high background concentration and low net
production; and Cluster 4 with high background concentration and high net production. The background concentration of ozone
is mainly determined by the nighttime loss of ozone and the titration of nitrogen oxides in the morning, while the net production
depends on the intensity of the photochemical reactions. In Figure 4b, the four ozone profiles occurred for 13, 24, 20, and 4 days
in 2020, comparable occupation of 19, 21, 17, and 4 days in 2021, respectively. During the static management period of 2022,
when the nitrogen oxide titration effect weakened, the number of days on which Cluster 3 and Cluster 4 appeared increased to 34
and 16, respectively. The average ozone levels for the four types were comparable across 2020, 2021, and 2022, ranging from





27-30 ppbv for Cluster 1, 38-40 ppbv for Cluster 2, 46-49 ppbv for Cluster 3, and 54-61 ppbv for Cluster 4 (Figure 4c). As depicted in Figure 4d, assuming that the proportions of the four types in 2022 are the same as those in 2020 and 2021, the average ozone levels in 2022 would decrease by 14.5% and 17.3%, respectively, remaining comparable to the levels in 2020 and 2021. Alternatively, if the proportions of the four types in 2020 and 2021 were the same as those in 2022, the average ozone levels in 2020 and 2021 would increase by 15.4% and 20.2%, respectively, which would be very close to the levels in 2022. Purely statistical analysis indicated that the significant increase in ozone levels in 2022 was due to a higher proportion of Cluster 3 and Cluster 4, which had higher ozone concentrations during the static management period.

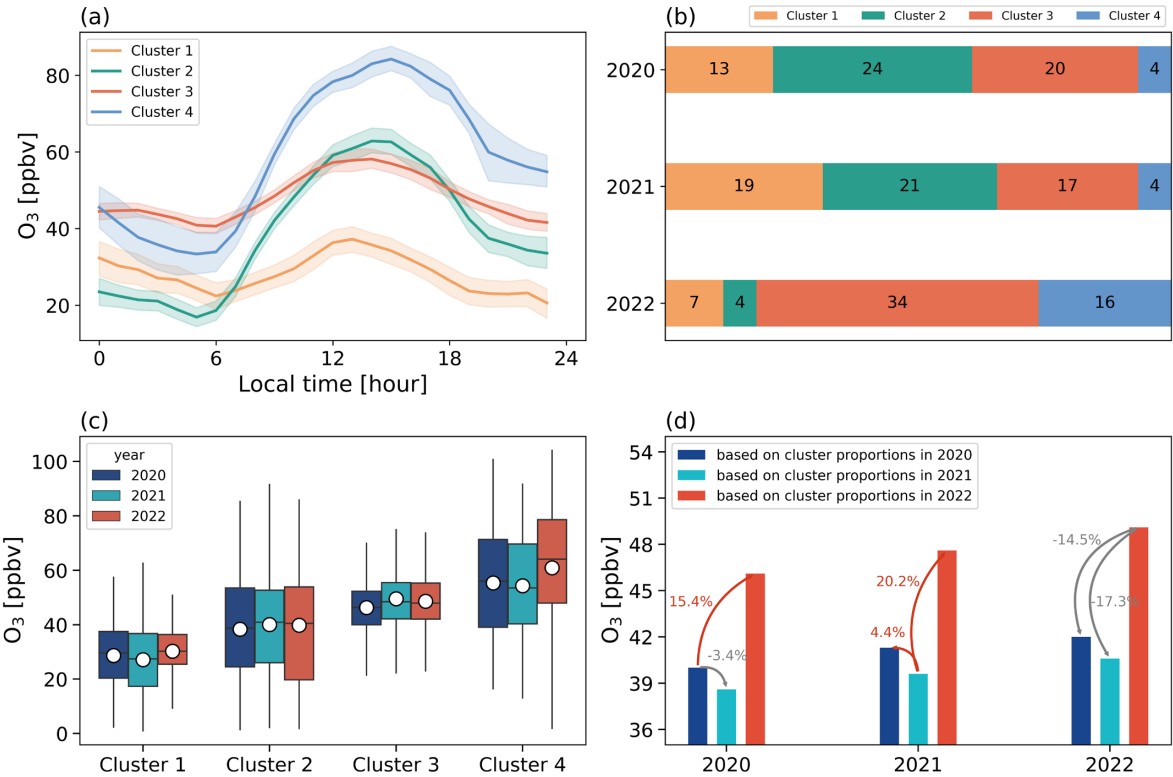

**Figure 4.** (a) Comparison of the mean diurnal profiles of the four types of O₃ after clustering. Colored areas denote 95% confidence intervals; (b) The proportions of the four clusters in 2020, 2021 and 2022. (c) Comparison of the O₃ levels of the four clusters in 2020, 2021 and 2022. The top and bottom of the vertical line for each box correspond to the 95th and 5th percentiles, respectively. The dots represent the averages, and the top, middle, and bottom lines of the box mark the 75th, 50th, and 25th percentiles, respectively; (d) Comparison of the average ozone concentrations in 2020, 2021 and 2022 for different ratios of the four clusters.

Different types of ozone profiles are formed under different meteorological conditions and pollution environments with distinctive diurnal variations. The meteorological conditions during the periods of Cluster 2 and Cluster 4, with high net ozone production, were characterized by high temperature, low relative humidity, and high radiation, compared to those during Cluster 1 and Cluster 3 periods. This is consistent with the well-known favorable condition promoting ozone production. The valley values of ozone profiles are closely related to the titration of nitrogen oxides, as shown in Figures 5f and 5h, where the valley values of ozone are inversely related to the peak values of NO₂ during the morning rush hour. Indeed, it is these meteorological conditions and titration that result in the formation of the corresponding four clusters of ozone profiles. During periods of Cluster 2 and Cluster 4, a large amount of VOCs accumulated before sunrise and were rapidly consumed after sunrise. Similarly, the precursor NO₂ was also rapidly consumed after sunrise, with the difference that the NO₂ level during Cluster 4 was lower than





that of Cluster 2. The VOCs/NO$_2$ ratio in Cluster 4 was significantly higher than that in other clusters, which may explain the
substantial net ozone production despite the relatively high ozone background levels. The differences among the clusters are also
reflected in the photochemical processes. In the following sections, we investigated the reasons for the increase in ozone levels
during the static management period from the perspective of atmospheric oxidizing capacity and free radical chemistry.

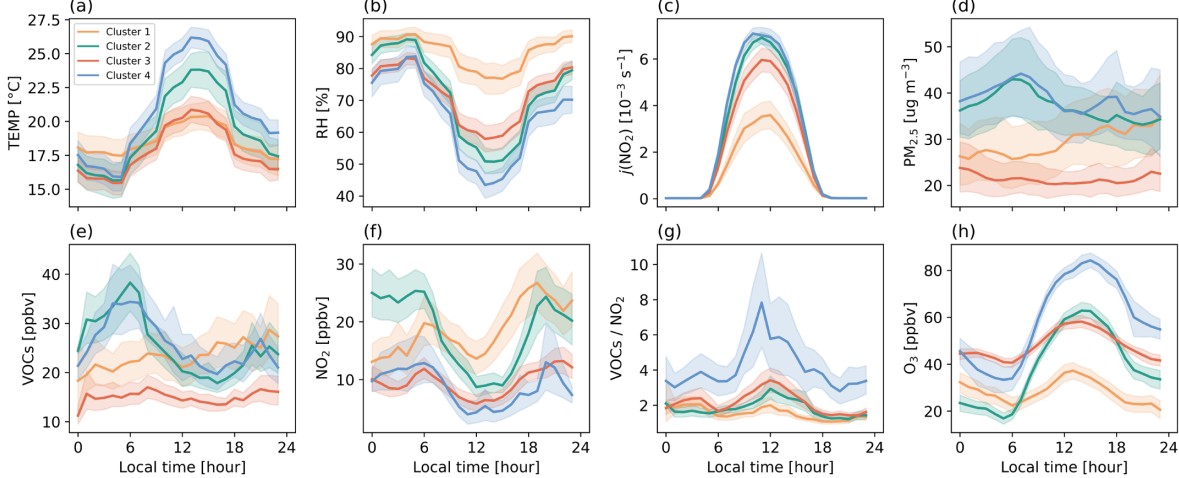


**Figure 5. The mean diurnal profiles of meteorological parameters and air pollutants for four clusters over three years. Colored areas**
**denote 95% confidence intervals.**
**3.3 Radical chemistry**
Figures 6a, 6e and 6i show the mean diurnal profiles of simulated OH, HO$_2$ and RO$_2$ radical concentrations for the years 2020,
2021, and 2022. These radicals exhibit clear diurnal variations, with peaks occurring at midday. The mean diurnal profiles
display that the average peak concentrations of OH were $4.0 \times 10^6$, $4.1 \times 10^6$, and $5.3 \times 10^6$, those of HO$_2$ were $2.0 \times 10^8$, $2.6 \times$
$10^8$, and $4.9 \times 10^8$ molecules cm$^{-3}$, and those of RO$_2$ were $0.8 \times 10^8$, $1.4 \times 10^8$ and $2.6 \times 10^8$ molecules cm$^{-3}$, respectively.
Reviewing previously observational results, peak concentrations of OH and HO$_2$ were observed at various locations and times:
$(4-17) \times 10^6$ molecules cm$^{-3}$ and $(2-24) \times 10^8$ molecules cm$^{-3}$ at a suburban site in Yufa from Aug 18-31, 2006 (Lu et al., 2013);
$(5-15) \times 10^6$ molecules cm$^{-3}$ and $(3-14) \times 10^8$ molecules cm$^{-3}$ at a rural site in Wangdu from June 8 to July 8, 2014 (Tan et al.,
2017); $4.5 \times 10^6$ molecules cm$^{-3}$ and $3 \times 10^8$ molecules cm$^{-3}$ at a suburban in Heshan from October 22 to November 5, 2014 (Tan
et al., 2019); $(2-9) \times 10^6$ molecules cm$^{-3}$ and $(2-14) \times 10^8$ molecules cm$^{-3}$ at a urban sites in Shenzhen from Oct 5-28, 2018 (Yang
et al., 2022); $(8-24) \times 10^6$ molecules cm$^{-3}$ and $(4-28) \times 10^8$ molecules cm$^{-3}$ at a suburban site in Taizhou from May 23 to June 18,
2018 (Ma et al., 2022); and $(10-20) \times 10^6$ molecules cm$^{-3}$ and $(6-18) \times 10^8$ molecules cm$^{-3}$ at a suburban site in Chengdu from
Aug 10-25, 2019 (Yang et al., 2021). The simulated concentrations of OH and HO$_2$ in this study were comparable to the
observed levels during autumn in Shenzhen, which is also an urban site, however, were generally lower than those observed at
non-urban sites. This difference can be attributed to the site types, but more importantly, to the fact that most observations were
conducted during periods of stronger radiation. For RO$_2$, the average maximum concentration was simulated to be $4.5 \times 10^8$
molecules cm$^{-3}$ at urban site of Beijing in August 2007. At coastal site of Xiamen, the simulated average daily peak reached 4.7
$\times 10^8$ molecules cm$^{-3}$ in September 2019, while at the coastal site of Ningde, the simulated value was $0.9 \times 10^8$ molecules cm$^{-3}$ in
spring 2019. Overall, our ROx concentrations fell within the range of observations and simulated results in other regions of
China. During the static management period in 2022, the levels of ROx were significantly higher compared to the same period in
2020 and 2021, indicating an enhanced atmospheric oxidation capacity in Shanghai in 2022.



Figures 6b-6d, Figures 6f-6h, and Figures 6j-6l illustrate the mean diurnal variation of primary OH, HO$_2$, and RO$_2$ sources for the years of 2020, 2021, and 2022. For OH, HONO photolysis peaked at around 07:00 and remained high until around 12:00, with peak values reaching approximately 0.57 ppbv h$^{-1}$, 0.41 ppbv h$^{-1}$, and 0.59 ppbv h$^{-1}$ in 2020, 2021, and 2022, respectively. Meanwhile, ozone photolysis peaked at noon, with peak values reaching around 0.65 ppbv h$^{-1}$, 0.74 ppbv h$^{-1}$, and 0.87 ppbv h$^{-1}$, respectively. In addition, the ozonolysis of unsaturated VOCs was another source of OH radical, with an average production rate of less than 0.10 ppbv h$^{-1}$, while other sources such as the photolysis of H$_2$O$_2$, HNO$_3$, and OVOCs were generally negligible. Overall, HONO photolysis for the daytime accounted for 57%, 42%, and 48% of the total OH primary production rates in 2020, 2021, and 2022, respectively, with O$_3$ photolysis accounting for 40%, 48%, and 47% in the corresponding years. For HO$_2$ radical, the most important source was HCHO photolysis, with average production rates during daytime of 0.20 ppbv h$^{-1}$, 0.17 ppbv h$^{-1}$, and 0.24 ppbv h$^{-1}$ in the corresponding years, respectively. The secondary source was OVOCs photolysis, which produce HO$_2$ at the rate of 0.10 ppbv h$^{-1}$, 0.15 ppbv h$^{-1}$, and 0.06 ppbv h$^{-1}$, respectively. Another source to consider was reactions of O$_3$ and unsaturated VOCs, which has an average rate of around 0.02 ppbv h$^{-1}$. For RO$_2$ radical, the daytime average peak of primary production rates contributed by OVOCs photolysis from 2020 to 2022 were 0.14 ppbv h$^{-1}$, 0.22 ppbv h$^{-1}$, and 0.10 ppbv h$^{-1}$, respectively. The reactions of O$_3$ and NO$_3$ with VOCs were also important primary sources of RO$_2$ radical. The average daily contributions of O$_3$+VOC reactions from 2020 to 2022 were 0.02 ppbv h$^{-1}$, 0.08 ppbv h$^{-1}$, and 0.04 ppbv h$^{-1}$, while the contributions of NO$_3$+VOC reactions were 0.08 ppbv h$^{-1}$, 0.13 ppbv h$^{-1}$, and 0.04 ppbv h$^{-1}$, respectively. The primary production rate of RO$_2$ in 2022 was lower compared to the years 2020 and 2021. This can be attributed to the fact that the primary sources of RO2 are the reactions involved VOCs, which significantly decreased during the static management period and further led to the decreased sournces of RO$_2$.

Overall, the primary production rates of ROx remained relatively stable from 2020 to 2022, with values of 1.20 ppbv h$^{-1}$, 1.35 ppbv h$^{-1}$, and 1.28 ppbv h$^{-1}$, respectively. These values are close to the simulated value of 1.56 ppbv h$^{-1}$ in November 2019 in downtown Shanghai (Zhang et al., 2022a), but lower than the value of 2.55 ppbv h$^{-1}$ during the ozone episode in the suburban of Shanghai in 2018 (Zhang et al., 2021). During these three years, reactions involving VOCs (excluding HCHO) accounted for 25%, 42%, and 19%, respectively, which is correlated with the observed VOCs abundances. In 2020 and 2022, HONO photolysis accounted for over 30% of the total primary production rates, followed by O$_3$ photolysis, which accounted for 23% and 30% respectively. In 2021, the dominant contribution was from O$_3$ photolysis, reaching 24%, followed by HONO photolysis at 20%. Radical chemistry exhibits heterogeneity across different cities, with HONO photolysis being a primary source in New York (Ren et al., 2003), Paris (Michoud et al., 2012), Wangdu (Tan et al., 2017), and Taizhou (Ma et al., 2022). OVOC photolysis dominated in Mexico City (Sheehy et al., 2010), Hong Kong (Xue et al., 2016), and Beijing (Liu et al., 2012). Milan relied on HCHO photolysis as a major source (Alicke et al., 2002), while ozone photolysis was prominent in Nashville (Martinez et al., 2003).



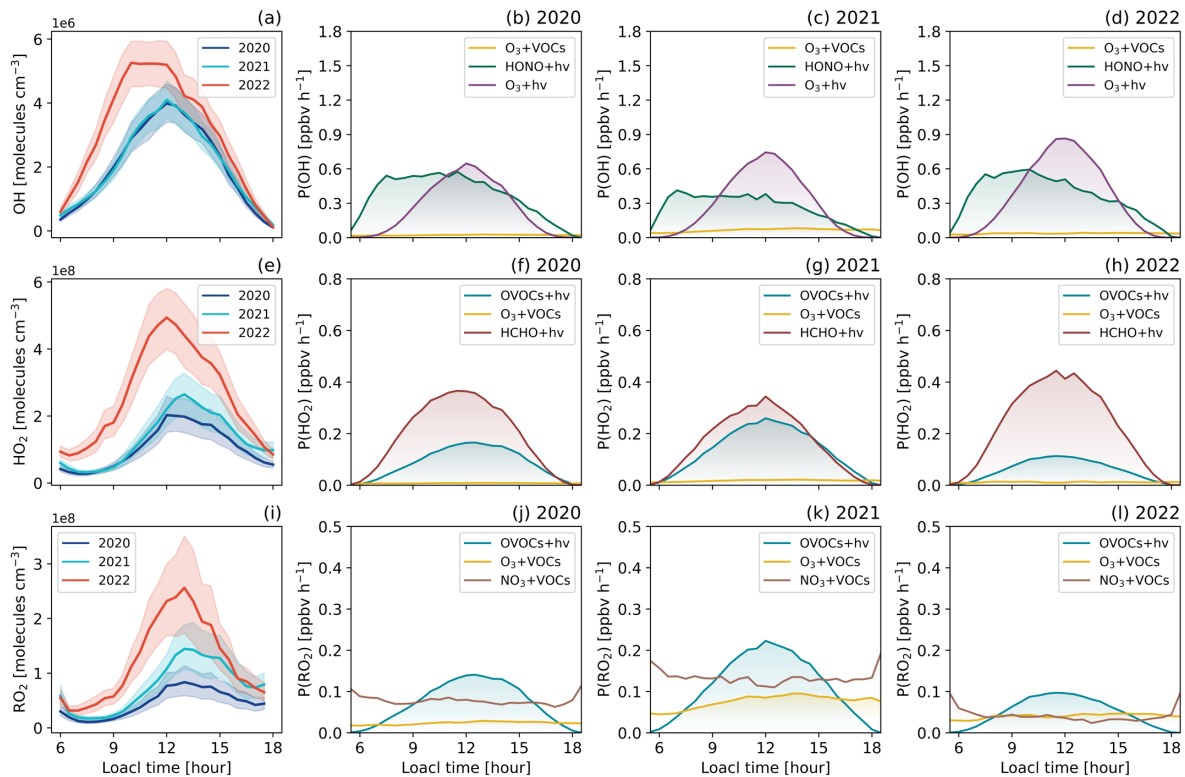

**Figure 6. The mean diurnal profiles of simulated OH (a), HO₂ (e), and RO₂ (i) concentrations in 2020, 2021, and 2022. Colored areas denote 95% confidence intervals; The mean diurnal profiles of primary sources of OH radical (b-d) , HO₂ radical (f-h), and RO₂ radical (j-l) from model calculations in 2020, 2021 and 2022.**

We also investigated the radical chemistry under different ozone profile clusters (Figure 7). The average peak of OH in Cluster 1, Cluster 2, Cluster 3, and Cluster 4 were $2.5 \times 10^6$ molecules cm³, $4.4 \times 10^6$ molecules cm⁻³, $4.8 \times 10^6$ molecules cm⁻³, and $6.1 \times 10^6$ molecules cm⁻³, those of HO₂ were $0.9 \times 10^8$ molecules cm⁻³, $2.5 \times 10^8$ molecules cm⁻³, $3.3 \times 10^8$ molecules cm⁻³, and $6.6 \times 10^8$ molecules cm⁻³, and those of RO₂ were $0.4 \times 10^8$ molecules cm⁻³, $1.3 \times 10^8$ molecules cm⁻³, $1.4 \times 10^8$ molecules cm⁻³, and $4.2 \times 10^8$ molecules cm⁻³, respectively. The order of those concentrations among the four clusters is consistent with the order of average ozone concentration. Cluster 2 and Cluster 4, characterized by significant net ozone production, exhibit distinct features in radical chemistry. The daily average of P(ROx) was 1.50 ppbv h⁻¹ in Cluster 2 and 1.89 ppbv h⁻¹ in Cluster 4, which is higher than the values of 0.78 ppbv h⁻¹ in Cluster 1 and 1.11 ppbv h⁻¹ in Cluster 3. In addition, HONO photolysis during the morning rush hour was particularly prominent in Cluster 2 and Cluster 4, with peak values reaching 0.67 ppbv h⁻¹ and 0.78 ppbv h⁻¹, respectively. In Cluster 2, HONO photolysis was the dominant source with a daily average of 0.40 ppbv h⁻¹, accounting for 26% of the total, and ozone photolysis followed with 0.34 ppbv h⁻¹, accounting for 23%. On the other hand, in Cluster 4, ozone photolysis taked the lead with 0.58 ppbv h⁻¹, representing 30% of the total, and HONO photolysis came next with 0.40 ppbv h⁻¹, accounting for 21%. Additionally, OVOCs photolysis (including HCHO) in Cluster 2 and Cluster 4 showed a significant increase compared to Cluster 1 and Cluster 3. In conclusion, a large amount of net ozone production implies the presence of active photochemical processes.




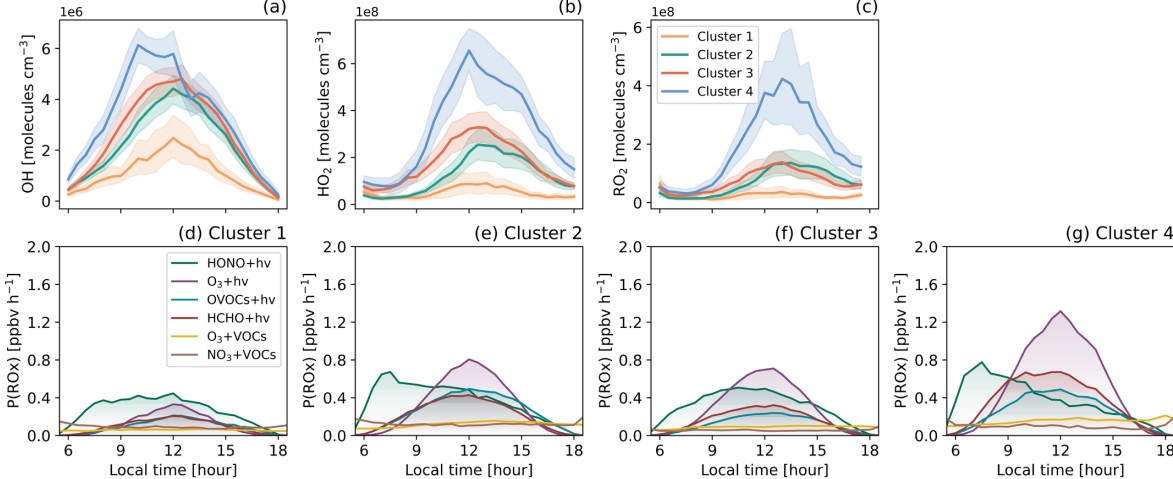

**Figure 7. The mean diurnal profiles of simulated OH (a), HO₂ (b), and RO₂ (c) concentrations for Cluster 1, Cluster 2, Cluster 3, and**
**Cluster 4. Colored areas denote 95% confidence intervals; The mean diurnal profiles of primary sources of ROx radical (d-g) from**
**model calculations for Cluster 1, Cluster 2, Cluster 3, and Cluster 4.**
In tropospheric chemistry, radical initiation, which involves the breakdown of closed-shell species to generate new radicals,
plays a crucial role in initiating the formation of secondary pollutants. However, in highly polluted atmospheric environments,
radical cycling becomes the dominant process, with the amplification of new radicals in the ROx cycle playing a crucial role.
Volkamer et al. (2010) quantified the production of new radicals and found that approximately 20% of radical production is
attributed to the breakdown of closed-shell species, while 80% is derived from radical cycling. Therefore, in addition to
understanding the sources of radicals, it is essential to comprehend the impacts of radical cycling and recycling processes on
ozone formation. Figure 8 demonstrated the daytime average of ROx radicals budgets during the periods from April to May of
2020, 2021, and 2022. Taking the simulation of 2022 as an example, OH oxidation of CO and VOCs produces HO₂ and RO₂
with daytime average rates of 2.11 ppbv h$^{-1}$ (1.44 ppbv h$^{-1}$ in 2020 and 1.45 ppbv h$^{-1}$ in 2021) and 1.07 ppbv h$^{-1}$ (0.78 ppbv h$^{-1}$ in
2020 and 1.17 ppbv h$^{-1}$ in 2021), respectively. The reactions of RO₂+NO and HO₂+NO further lead to the strong production of
RO with a rate of 1.13 ppbv h$^{-1}$ (0.98 ppbv h$^{-1}$ in 2020 and 1.51 ppbv h$^{-1}$ in 2021) and OH with a rate of 3.35 ppbv h$^{-1}$ (2.76 ppbv
h$^{-1}$ in 2020 and 3.22 ppbv h$^{-1}$ in 2021), while generating O₃ as a by-product. Clearly, these recycling processes dominate the
overall production of ROx radicals compared to the primary sources. In terms of termination processes, the loss of ROx radicals
was primarily dominated by their reactions with NOx. Specifically, the reactions of OH+NO₂ and RO₂+NO₂ accounted for
approximately 0.51 ppbv h$^{-1}$ (0.76 ppbv h$^{-1}$ in 2020 and 0.74 ppbv h$^{-1}$ in 2021) and 0.80 ppbv h$^{-1}$ (0.96 ppbv h$^{-1}$ in 2020 and 1.55
ppbv h$^{-1}$ in 2021) of the ROx radical loss on daytime average, respectively. This is in line with the understanding that reactions
involving NOx typically play a dominant role in the removal of radicals in high NOx environments (Zhang et al., 2021; Xue et
al., 2016; Volkamer et al., 2010; Tan et al., 2019; Liu et al., 2012). To sum up, the changes resulting from the approximately 55%
reduction in NO₂ and 30% reduction in VOCs due to static management are reflected in both the radical propagation and
termination processes. The ratio of OH radical propagation (OH+VOCs) to termination (OH+NO₂) reached 2.10, which is higher
than 1.03 in 2020 and 1.60 in 2021. It can be interfered the different proportions of NO₂ and VOCs reduction did not weaken the
radical cycling. On the contrary, a higher VOCs/NO₂ ratio promotes the radical recycling efficiency in the reaction chain of
radicals.



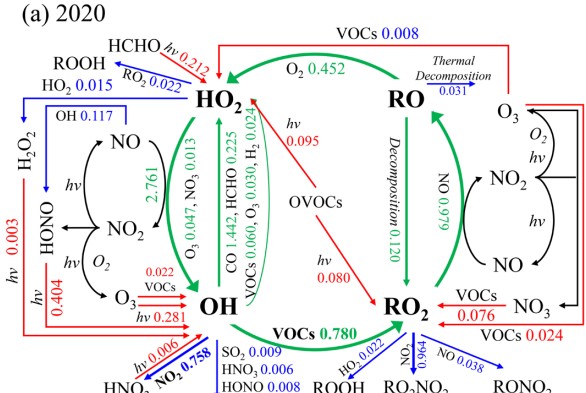

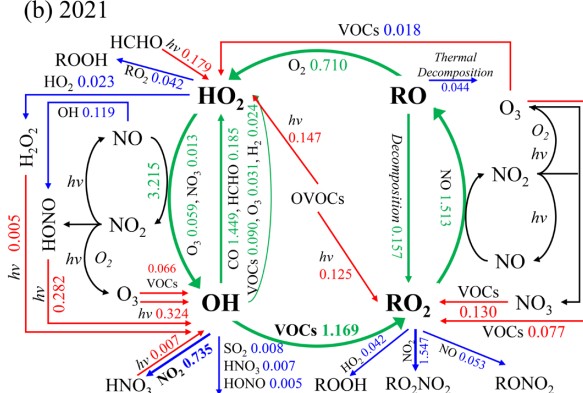

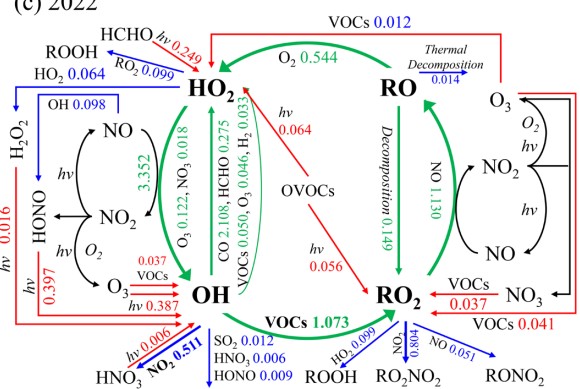

**Figure 8. Averaged budgets (in ppbv h$^{-1}$) of ROx radicals in daytime (06:00-18:00) during the periods from April to May of 2020, 2021, and 2022. The red, blue, and green lines and words indicate the primary production, termination, and recycling pathways of radicals, respectively.**

## 4 Conclusions

The two-month city-wide static management was implemented in April and May 2022 in Shanghai aiming to control the spread of the Omicron variant, which provides a valuable opportunity to study the causes of ozone pollution. The comprehensive



observations during the static management and same period in 2020 and 2021 shows that , there was a decrease of 29%-35% in VOCs and 51%-55% in $NO_2$ concentrations, while the average ozone level increased by nearly 23%. By statistics, the ozone profiles were classified into four clusters: Cluster 1, characterized by low background concentration and low net production; Cluster 2, characterized by low background concentration and high net production; Cluster 3, characterized by high background concentration and low net production; and Cluster 4, characterized by high background concentration and high net production. The average concentration relationship among these clusters is Cluster 4 > Cluster 3 > Cluster 2 > Cluster 1. The significant increase in the proportion of Cluster 4 and Cluster 3 during the period of static management resulted in the overall increase in the average ozone level. Secondly, from the perspective of radical chemistry, we explored the changes in photochemical processes due to the reduction in precursor species. The OBM model simulated the levels of radicals and their processes of initiation, propagation, and termination. The average peak concentrations of OH, $HO_2$, and $RO_2$ in 2022 were $5.3 \times 10^6$, $4.9 \times 10^8$, and $2.6 \times 10^8$ molecules $cm^{-3}$, respectively, which were higher than those in the same period in 2020 and 2021. HONO photolysis was the main contributor to the primary source of ROx, accounting for about 30% of the total. However, in terms of the overall production of ROx radicals, the radical recycling process remained dominant. The reduction of $NO_2$ and VOCs in different proportions due to static management has led to an increased ratio of OH radical propagation (OH+VOCs) to termination (OH+$NO_2$), reaching 2.10, which is higher than 1.03 in 2020 and 1.60 in 2021. This enhanced ratio indicates a strengthened radical cycling as a result of a higher VOCs/$NO_2$ ratio.

The important lesson we learned from the "large-scale field experiment" during the period of static management is that Shanghai is in VOCs-limited regime. When the reduction in VOCs is not able to catch up with or exceed the reduction in nitrogen oxides, it is not sufficient to curb the formation of secondary pollutants. In terms of ozone control strategies, it is necessary to strengthen the regulation and control of VOCs.

**Data Availability**

The observed and predicted hourly time series data in the study are presented in Figure S11-S12, and Code and data used for our analyses are available at *https://data.mendeley.com/datasets/3kmhg7r2df/1*(Zhu, 2023).

**Competing interests**

The contact author has declared that none of the authors has any competing interests.

**Author contributions**

Jian Zhu: Conceptualization, Methodology, Software, Validation, Investigation, Writing original draft, Visualization. Shanshan Wang: Conceptualization, Methodology, Supervision, Funding acquisition. Chuanqi Gu, Zhiwen Jiang, Sanbao Zhang, Ruibin Xue, and Yuhao Yan: Methodology, Validation, Investigation. Bin Zhou: Conceptualization, Methodology, Supervision, Funding acquisition.

**Acknowledgments**

This work was supported by National Natural Science Foundation of China (22176037, 42075097, 22376030, 42375089, 21976031) and National Key Research and Development Program of China(2022YFC3700101).



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
