# Peer review of "Why Did Ozone Concentrations Remain High During Shanghai's Static 2 Management? A Statistical and Radical Chemistry Perspective"

_EGUsphere, 2023_

## Author Comment (AC1)

General comments:

The article investigates the increase in ozone levels in Shanghai during city-wide static management measures implemented in April and May 2022 to control the spread of the Omicron variant. Despite significant reductions in VOCs and NO2 emissions, average ozone levels rose by 23%. The study involves comprehensive observations and statistical analysis of ozone profiles categorized into four clusters based on their characteristics. It highlights the importance of radical chemistry and photochemical processes in understanding the increased ozone levels. The research concludes that Shanghai operates in a VOCs-limited regime, suggesting that controlling VOCs is vital for effective ozone pollution management. I generally agree with the structure of the manuscript and classify my recommendation as minor revision. However, there are several specific and minor points that I recommend the authors address (see below).

Firstly, we would like to express our sincere gratitude for the valuable time and effort you have dedicated to reviewing my manuscript. We have also re-examined the article and identified issues with the processing of photolysis data. Consequently, we have reran the OBM model and conducted a fresh statistical analysis and discussion of the results related to the OBM model (Section 3.3). The latest findings have not fundamentally altered the original conclusions. The responses to your comments that follow are based on these most recent results.

Specific comments:

How were the simulated radical concentrations validated against observed data, especially considering the variations in different urban and non-urban environments?

R: Thanks for your comments. Unfortunately, this study did not conduct simultaneous observations of free radicals, thus direct verification was not possible. However, there are research teams in China that have carried out comparative studies on radicals' observations and simulations.

The earliest observation of OH radicals in China was accomplished through collaborative efforts between Peking University and Forschungszentrum Juelich (FZJ), resulting in the construction of a Laser Induced Fluorescence (LIF) instrument for detecting OH and $HO_2$. Additionally, the Anhui Institute of Optics and Fine Mechanics (AIOFM) independently developed a Laser Induced Fluorescence (LIF) instrument. Simultaneously, comparisons with model simulations were conducted during the radical observations.

Tan et al. (2017) from Peking University's observations and simulation studies in Wangdu, China, indicate that the simulation of radicals exhibits excellent performance. Zhang et al. (2022a) from AIOFM reported good consistency between observed and simulated values of radicals. Those indicate the feasibility of applying observation-based models in the simulation of photochemical processes. Based on error analysis of models and observations, Lu and Zhang (2010) suggested that the ratio of simulated to measured values for HOx (OH + $HO_2$) falling within the range of 0.4-1.6 can be considered acceptable. Applying the chemical box model to dissect photochemical processes is a mature and widely-used approach, currently adopted by teams that lack the capability to conduct radical observations.

Lu et al. (2019) categorized the observed OH radicals into three groups: urban areas, remote

regions (including continents, oceans, polar regions, and the free troposphere), and forested areas. The maximum concentrations of OH radicals observed across different categories all fall within the range of $10^6$ to $10^7$ molecules $cm^{-3}$ (see Figure R1). In this study, the average peak of OH was around $5.0 \times 10^8$ molecules $cm^{-3}$, which is considered a moderate level in urban environments.

The team from AIOFM conducted continuous observations of radicals at the Shanghai Academy of Environmental Sciences (31.17°N, 121.43°E, urban environment) using their independently developed Laser-Induced Fluorescence (LIF) system from November 8th to November 24th, 2019 (Zhang et al., 2022a). The average concentrations of OH and $HO_2$ radicals during the midday hours (11:00-13:00) were $2.7 \times 10^6$ and $0.8 \times 10^8$ molecules $cm^{-3}$, respectively. We performed OBM simulations for the same period at the site within this study. The simulation results showed average concentrations of OH and $HO_2$ radicals to be $2.67 \times 10^6$ and $0.84 \times 10^8$ molecules $cm^{-3}$, respectively. The diurnal variation trends also demonstrated good consistency. This indirectly indicates that the free radical chemistry simulation results of this study are relatively reliable.

We hope to conduct radical observations in the future through collaborative efforts or independent research, and to conduct comparative studies on radicals in urban and non-urban environments.

[Figure]

**Figure R1. Typical observed daily averaged maximum OH concentrations at distinct different geophysical regions (i.e. urban, remote, forest areas) with different measurement techniques (i.e. DOAS, LIF, CIMS, CEAS and CRDS). The figure is sourced from Lu et al. (2019).**

The study details various sources and production rates for radicals. If possible, a deeper exploration of how these sources differ in urban settings and their specific impacts on air quality could be insightful.

R: Thanks for your comments. Over the past few decades, understanding the sources and chemical behavior of HOx has been a focal point in air quality research. Despite the visual similarities in air pollution issues, the chemistry of radicals, particularly the relative importance of the main sources of radicals, varies across different types of regions. Generally, the primary sources of radicals include the photolysis of $O_3$, HONO, HCHO, and other OVOCs, as well as non-photolytic sources such as the reaction of ozone with alkenes and the reaction of $NO_3$ with unsaturated VOCs.

Regarding the impact of the sources of radicals in urban environments on air quality, we conducted a correlation analysis between the sources of radicals and ozone-related indices, as shown in Figure R2. The production rates of radicals P(ROx) had a strong positive correlation with the production rates of ozone P($O_3$), and both were temperature-dependent. The higher the temperature, the higher P($O_3$) and P(ROx). Additionally, we defined daily net ozone production

O$_3$_net as the difference between the highest ozone value at noon and the lowest value in the morning. The correlation analysis between daily O$_3$_net and the photolysis rates of HONO, O$_3$, HCHO, and OVOCs, as well as the total radical source, reveals a significant positive relationship. This implies that the strength of different sources of modeled radicals positively contributes to ozone production. From the perspective of the slopes, the contribution of HCHO photolysis to net ozone generation is evident. This may be attributed to the enhancement of the NO$_2$-NO-O$_3$ cycle by the HO$_2$ radicals produced from HCHO photolysis.

[Figure]

**Figure R2. Correlation analysis between the sources of radicals and ozone-related indices, (a) P(O$_3$) & P(ROx); (b) O$_3$_net & HONO+hv; (c) O$_3$_net & O$_3$+hv; (d) O$_3$_net & HCHO+hv; (e) O$_3$_net & HONO+hv; (f) O$_3$_net & P(ROx).**

We have included the above textual and graphical descriptions in the Supplement, please refer to Text S2 and Figure S15. Further explanations can be found in the manuscript on Lines 310-311.

We have also included in the Supplement a review of the radical observation and simulation studies conducted both domestically and internationally, as shown in the following text and table. Please refer to Text S3 and Table S3 in the Supplement.

"*We have compiled observations and simulations of radical chemistry conducted by various research teams in different environments (see Table R1). It is evident that the primary sources of radicals vary in different regions and seasons. Compared to marine boundary layer environments, forest environments, and suburban environments, the primary sources of radicals in urban environments were complex. For example, simulation work conducted in urban environments in Nashville, Houston, and Writtle indicates that the photolysis of ozone was a primary contributor to the primary source of radicals (Kovacs et al., 2003; Martinez et al., 2003; Thornton et al., 2002; Mao et al., 2010; Lee et al., 2006; Emmerson et al., 2007; Emmerson and Carslaw, 2009). And in locations like New York, Palaiseau, Shenzhen, the photolysis of HONO played a significant role (Ren et al., 2003a; Ren et al., 2003b; Cai et al., 2008; Dolgorouky et al., 2012; Michoud et al., 2012; Yang et al., 2022). It is noteworthy that within the same urban environment, the dominant sources can vary across different seasons, time periods, and conditions. In Tokyo, the primary sources of free radicals in winter were the reactions between ozone and alkenes, while in summer,*

*they were the photolysis of ozone and OVOCs (Kanaya et al., 2008; Kanaya et al., 2007a). In New York, during the daytime, the primary source was the photolysis of HONO, while at night, it resulted from the reaction between $O_3$ and alkenes(Cai et al., 2008; Ren et al., 2006). In Jülich, under high NOx conditions, the primary source was the photolysis of HONO, whereas under low NOx conditions, it was the photolysis of $O_3$ (Kanaya et al., 2012). In marine boundary layer environments, most studies indicated that the photolysis of ozone was the primary contributor to free radicals during the day, while at night, it resulted from the reaction between $O_3$ and alkenes. In suburban and rural environments, the primary contributor was the photolysis of HONO. Just as you mentioned the differences in radical chemistry between urban and suburban environments, historical research has also indicated this, and it is a crucial aspect of the simulation studies we conducted in Shanghai.*"

**Table R1. Summary of measurements and model comparisons for OH and HO$_2$ in different typical environments, as well as OH reactivity and HOx primary sources**

| campaign | year | location | Technique | OH measured | HO$_2$ measured | OH reactivity | HOx primary sources | comments | Ref. |
|---|---|---|---|---|---|---|---|---|---|
| **Urban environment** | | | | | | | | | |
| SOS | June-July 1999 | Nashville, Tennessee, USA, 36°N, 88°W | LIF | 0.8 pptv at noon | 80 pptv at noon | 11.3±4.8 s$^{-1}$ | O$_3$+hv | OH observed-to-modeled ratio: 1.33; OH observed-to-modeled ratio: 1.56; the measured OH reactivity is about 1.2 times larger than modeled. | (Thornton et al., 2002; Martinez et al., 2003; Kovacs et al., 2003) |
| PUMA | June 1999 & January-February 2000 | Birmingham, UK, 53°N, 2°W | LIF | (2-9)×10$^6$ molecule cm$^{-3}$ in summer, (0.5-4)×10$^6$ molecule cm$^{-3}$ in summer, | (1.5-10)×10$^8$ molecule cm$^{-3}$ in summer, ~4×10$^6$ molecule cm$^{-3}$ in summer, | - | O$_3$+alkenes: 46% in summer, 62% in winter | OH modeled-to-measured ratio: 0.58 in summer and 0.5 in winter; HO$_2$ modeled-to-measured ratio: 0.56 in summer and 0.49 in winter. | (Emmerson et al., 2005a; Emmerson et al., 2005b; Heard et al., 2004; Harrison et al., 2006) |
| TEXAQS | August-September 2000 | Houston, Texas, USA, 29°N, 95°W | LIF | Maximum ~0.8 pptv | Maximum ~30 pptv | 7-12 s$^{-1}$ | O$_3$+hv | | (Mao et al., 2010) |
| PMTACS-NY | June-August 2001 | New York, USA, 41°N, 74°W | LIF | (2-20)×10$^6$ molecule cm$^{-3}$ | (0.5-2)×10$^8$ molecule cm$^{-3}$ | 15-25 s$^{-1}$ | HONO+hv | OH observed-to-modeled ratio: 1.10; HO$_2$ observed-to-modeled ratio: 1.24; The OH reactivity measurements agree with the calculations to within 10% | (Cai et al., 2008; Ren et al., 2003b; Ren et al., 2003a) |
| HOxComp | 9-11 July 2005 | Julich, Germany (50°54′33″N, 06°24′44″E) | LIF, DOAS, CIMS | Maximum 9.4×10$^6$ molecule cm$^{-3}$ | Maximum 35 pptv | | high-NOx: HONO+hv; low-NOx: O$_3$+hv | Good agreement between model and observations for OH under high isoprene low NOx (fresh isoprene emissions) | (Kanaya et al., 2012) |
| MCMA | April 2003 | Mexico City, Mexico, 19°N,100°W | LIF | (5-8)×10$^6$ molecule cm$^{-3}$ | 15-60 pptv | 25-120 s$^{-1}$ | HCHO+hv | (OH)model=1.2×(OH)-0.008 pptv R$^2$=0.80; (HO$_2$)model=0.90×(HO$_2$)+0.98 pptv R$^2$=0.64; OH reactivity in the model is about 10-20% too low | (Volkamer et al., 2010; Sheehy et al., 2010; Shirley et al., 2006) |
| TORCH-1 | July-August 2003 | Writtle, Essex, | LIF | (1.2-7.5)×10$^6$ molecule cm$^{-3}$, with | (0.16-3.3)×10$^8$ molecule cm$^{-3}$, | 2-10 s$^{-1}$ | O$_3$+hv | Daytime OH and HO$_2$ Overpredicted by 24% and 7%, respectively; nighttime | (Emmerson and Carslaw, 2009; |

| | | | | | | | | |
|---|---|---|---|---|---|---|---|---|
| | | UK, 51°N, 0°E | | $2.6\times10^6$ molecule $cm^{-3}$ at night | with $2.9\times10^7$ molecule $cm^{-3}$ at night | | | OH and $HO_2$ unpredicted 41% and 16%, respectively | (Emmerson et al., 2007; Lee et al., 2006) |
| IMPACT IV&L | January-February and July-August 2004 | Tokyo, Japan, 35°N, 139°E | LIF | In winter, daytime median was $1.5\times10^6$ molecule $cm^{-3}$ and night mean was $1.8\times10^5$ molecule $cm^{-3}$; in summer, daytime median was $6.3\times10^6$ molecule $cm^{-3}$ and night mean was $3.7\times10^5$ molecule $cm^{-3}$ | In winter, daytime median was 1.1 pptv and night mean was 0.7 pptv; in summer, daytime median was 5.7 pptv and night mean was 2.6 pptv | | the day and night in winter: $O_3$+alkenes; in daytime during summer: $O_3$+hv and OVOCs+hv, during the early morning in summer: HONO+hv | Daytime OH well reproduced by model for both periods; daytime $HO_2$ underestimated in winter and overestimated in summer. | (Kanaya et al., 2008; Kanaya et al., 2007a) |
| PMTACS-2 | January-February 2004 | New York, USA, 41°N, 74°W | LIF | maximum $1.4\times10^6$ molecule $cm^{-3}$ | maximum 0.7 pptv | 20-40 $s^{-1}$ | Daytime: HONO+hv; Nighttime: $O_3$+alkenes | OH observed to modelled ratio of 0.98; $HO_2$ observed to modelled ratio of 6, with the greatest values at high NO; OH reactivity measured higher than calculated at rush hour in the morning and in the evening, possibly due to unmeasured or missing VOCs. | (Cai et al., 2008; Ren et al., 2006) |
| MILAGRO | March 2006 | Mexico City, Mexico, 19°N, 100°W | LIF | maximum median $4.6\times10^6$ molecule $cm^{-3}$ | maximum median $1.9\times10^8$ molecule $cm^{-3}$ | | HCHO+hv | OH overpredicted by a factor of 1.7 at midday, well reproduced after 14:30; $HO_2$ underpredicted in the morning, well reproduced after 11:30 | (Dusanter et al., 2009b; Dusanter et al., 2009a) |
| TRAMP | August-September 2006 | Houston, Texas, USA, 29°N, 95°W | LIF | Daytime: 0.33±0.23 pptv; Nighttime: 0.087±0.066 pptv | Daytime: 22±18 pptv; Nighttime: 11±7.8 pptv | 10-20 $s^{-1}$ | morning rush hour: HONO+hv; daytime: $O_3$+hv; nighttime: $O_3$+alkenes | the measured OH was generally greater than the modeled OH for all mechanisms, especially during the afternoon; For $HO_2$, Good agreement was found in the morning for the models with most mechanisms, except for SAPRC-99 (slope=0.67); This good agreement between measured and calculated OH reactivity | (Chen et al., 2010; Mao et al., 2010) |
| MEGAPOLI | July 2009 | Palaiseau, French 48.71°N, 2.21°E | LIF | $\sim5\times10^6$ molecule $cm^{-3}$ | $\sim1.2\times10^8$ molecule $cm^{-3}$ | | HONO+hv | Photo Stationary State calculations overestimate OH by 50 %, box model overestimated 12% and 5 %, for OH and ($HO_2$+$RO_2$) respectively. | (Michoud et al., 2012; Dolgorouky et al., 2012) |

| | | | | | | | | |
|---|---|---|---|---|---|---|---|---|
| STORM | September-October 2018 | Peking University Shenzhen Graduate School, Shenzhen, China 22.60°N, 113.97°E | LIF | maximum of 4.5×10⁶ molecule cm⁻³ | Maximum of 4.2×10⁸ molecule cm⁻³ | 18-22 s⁻¹ | HONO+hv | Good agreement between the observed and modeled kOH during the several days in Shenzhen | (Yang et al., 2022) |

**Marine boundary environment**

| | | | | | | | | |
|---|---|---|---|---|---|---|---|---|
| EASE97 | April-May 1997 | Mace Head, Ireland, 53°N, 10°W | LIF | (2.0-6.0)×10⁶ molecule cm⁻³ | (0.5-3.5)×10⁸ molecule cm⁻³ | O₃+hv | average model-measurement ratios were 2.4 for OH, 3.6 for HO₂ between 11:00 and 15:00. | (Creasey et al., 2002; Carslaw et al., 2002) |
| OKIPEX | July-August 1998 | Oki Dogo Island, Japan, 36°N, 133°E | LIF | Below the detection limit of the instrument (0.8 pptv) | Maximum of 17 pptv | O₃+hv | Model usually overestimated HO₂ by a factor of 2 | (Kanaya et al., 2000; Kanaya and Akimoto, 2002) |
| ORION99 | August 1999 | Cape Hedo, Okinawa Island, Japan, 27°N, 128°E | LIF | Maximum of 4×10⁶ molecule cm⁻³ | Daytime: maximum of 17 pptv; nighttime: 0.5-5.5 pptv | O₃+hv | OH was underestimated by model calculations but within a large uncertainty; Model underestimated daytime HO₂ by only 20% | (Kanaya et al., 2001b; Kanaya et al., 2001a; Kanaya et al., 2002a) |
| SOAPEX-2 | January-February 1999 | Cape Grim, Tasmania, 41°S, 142°E | LIF | Maximum of 3.5×10⁶ molecule cm⁻³ | Maximum of 2×10⁸ molecule cm⁻³ | O₃+hv | Models overestimated OH by 10%-20%; Models overestimated HO₂ by ~40%; | (Sommariva et al., 2004; Creasey et al., 2003) |
| RISOTTO | June 2000 | Rishiri Island, Japan, 45°N, 141°E | LIF | | ~10 pptv at midday; mean of 4.2±1.2 pptv at night | Nighttime: O₃+alkenes | OH overestimated by ~36%; HO₂ overestimated by ~70%, requiring 25 pptv IO to reconcile model with observations | (Kanaya et al., 2002c; Kanaya et al., 2002b) |
| RISFEX | September 2003 | Rishiri Island, Japan, 45.07°N, 141.12°E | LIF | Daytime: maximum of 2.7×10⁶ molecule cm⁻³; Nighttime: (0.07-0.55)×10⁶ molecule cm⁻³ | Daytime: maximum of 5.9 pptv; Nighttime: (0.5-4.9) pptv | Daytime: O₃+hv; Nighttime: O₃+alkenes | OH overestimated by 35%; HO₂ overestimated by 89%; median nighttime modeled-to-observed ratios were 1.29 and 0.56 for HO₂ and OH, respectively | (Kanaya et al., 2007b; Qi et al., 2007) |
| RHaMBLe | May-June 2007 | Cape Verde, Atlantic Ocean, 16.85°N, 24.87°W | LIF | Maximum of 9×10⁶ molecule cm⁻³ | Daytime: maximum of 6×10⁸ molecule cm⁻³; Nighttime: ~0.6 pptv | Daytime: O₃+hv; Nighttime: O₃+alkenes | under-predicted OH on average by 18%; under-predicted HO₂ by 39% | (Lee et al., 2010; Whalley et al., 2010) |
| SOS | February-March; June; | Cape Verde, Atlantic Ocean, | LIF | Maximum of ~9×10⁶ | maximum of 4×10⁸ molecule | O₃+hv | Concentrations in summer (June, | (Carpenter et al., 2010; Vaughan et |

| September 2009 | 17°N, 25°W | | molecule cm$^{-3}$ | cm$^{-3}$, ~10$^7$ molecule cm$^{-3}$ at night | | | September) almost double those observed in winter (Feb, March) | al., 2012) |

**Forest environment**

| | | | | | | | | | |
|---|---|---|---|---|---|---|---|---|---|
| AEROBIC | July-August 1997 | Indigenous forest, North West Greece, 40°N, 21°E | LIF | (4-12)×10$^6$ molecule cm$^{-3}$ | (0.4-9)×10$^8$ molecule cm$^{-3}$ | | O$_3$+hv | Model underprediction of OH by ~50%. Modelled HO$_2$ was typically higher than observations. But observations showed high variability | (Creasey et al., 2001; Carslaw et al., 2001) |
| PROPHET-98 | August 1998 | Deciduous forest, North Michigan, USA, 45.6°N, 84.7°W | LIF | Daytime: 0.1-0.2 pptv; Nighttime: 0.04 pptv | Daytime: 10-25 pptv: Nighttime: 1-4 pptv | | O$_3$+alkenes | OH observations 2.7 times greater than the model; HO$_2$ observations and model in good agreement | (Faloona et al., 2001; Tan et al., 2001) |
| GABRIEL | October 2005 | Suriname, South America | LIF | 0.25 pptv | ~(50-55) pptv | | O$_3$+hv | Observed to modelled ratio of 12.2 for OH and 4.1 for HO$_2$ | (Kubistin et al., 2010; Butler et al., 2008; Lelieveld et al., 2008; Martinez et al., 2010) |
| OP3 | April and July 2008 | Sabah, Borneo, 5°N, 118°E | LIF | 2.5×10$^6$ molecule cm$^{-3}$ | 2×10$^8$ molecule cm$^-$ | 10-60 s$^{-1}$ | O$_3$+hv | Factor of 10 underprediction in OH when model constrained to OH reactivity. HO$_2$ overpredicted at ground level | (Whalley et al., 2011; Stone et al., 2011; Pugh et al., 2010) |
| HUMPPA-COPEC-2010 | July and August 2010 | boreal forest in Hyytiälä, southern Finland 61.9°N, 24.3°E | LIF, CIMS | 3.5×10$^6$ molecule cm$^{-3}$ | 37 pptv | 12.4 s$^{-1}$ | O3+hv | OH$_{LIF}$/OH$_{CIMS}$ = (1.31 ± 0.14); OH$_{mod.}$/OH$_{obs.}$=1.00 ± 0.16; HO$_2^{mod.}$ /HO$_2^{obs}$ = 0.3 ± 0.2; simulated OH reactivity does not match the observed OH reactivity | (Hens et al., 2014; Nölscher et al., 2012) |

**Suburban & rural environments**

| | | | | | | | | | |
|---|---|---|---|---|---|---|---|---|---|
| CAREBeijing-2006 | August- early September 2006 | Suburban rural site, Beijing, China, 39.61°N, 116.30°E | LIF | (4-17)×10$^6$ molecule cm$^{-3}$ | (2-24)×10$^8$ molecule cm$^{-3}$ | 10-30 s$^{-1}$ | HCHO+hv | A large discrepancy of a factor 2.6 is found at the lowest NO concentration encountered (0.1 ppb) | (Lu et al., 2010) |
| PRIDE-PRD | autumn 2014 | Guangdong Atmospheric | LIF | maximum median of 4.5×10$^6$ molecule cm$^-$ | maximum median of | 22-32 s$^{-1}$ | HONO+hv | 50 % was unexplained by the measured | (Tan et al., 2019) |

| | Date | Location | Method | OH | HO$_2$ | Reactivity | Source | Results | References |
|---|---|---|---|---|---|---|---|---|---|
| 2014 | | Supersite, China 22.73°N, 112.93°E | | [3] | 3×10$^8$ molecule cm$^{-3}$ | | | OH reactants | |
| | June 2014 | Wangdu, China | LIF | (5-15)×10$^6$ molecule cm$^{-3}$ | (3.14)×10$^8$ molecule cm$^{-3}$ | 10-20 s$^{-1}$ | HONO+hv | Model-measurement ratio was between 1.4 and 2; For HO$_2$, good agreement between modeled and observed concentrations during day and night | (Fuchs et al., 2017; Tan et al., 2017) |
| BEST-ONE | January-March 2016 | suburban site Huairou, Beijing, China 40.41°N, 116.68°E | LIF | 2.4×10$^6$ molecule cm$^{-3}$ in severely polluted air; 3.6×10$^6$ molecule cm$^{-3}$ in relatively clean air | 0.52×10$^8$ molecule cm$^{-3}$ in severely polluted air; 0.93×10$^8$ molecule cm$^{-3}$ in relatively clean air | 26.9 s$^{-1}$ in severely polluted air; 10.1 in relatively clean air | HONO+hv | OH and HO$_2$ observed to modelled ratio of 1.5 during clean days; underestimated HO$_2$ concentrations by factors up to 5 during pollution episodes | (Tan et al., 2018) |
| CHOOSE-2019 | August-September 2019 | Xinjin, Chengdu, China 30.40°N, 103.85°E | LIF | Mean of 9.5×10$^6$ molecule cm$^{-3}$ | Mean of 9×10$^8$ molecule cm$^{-3}$ | | HONO+hv | OH observed to modelled ratio of 0.8, HO$_2$ observed to modelled ratio of 1.0 | (Yang et al., 2021; Zhang et al., 2022b) |

The study relates different ozone profile clusters to radical chemistry variations. Further analysis of how these clusters specifically influence overall ozone levels and atmospheric chemistry would be beneficial.

R: Thanks for your comments.

Firstly, cluster analysis reveals that the ozone profiles in the Shanghai region can be classified into four clusters, each characterized by: Cluster 1 with low background concentration and low net production; Cluster 2 with low background concentration and high net production; Cluster 3 with high background concentration and low net production; and Cluster 4 with high background concentration and high net production, as shown in Table R2.

The common characteristic of Cluster 1 and Cluster 2 is a lower baseline value of ozone, primarily because a significant amount of ozone was titrated and consumed from nighttime to early morning. As depicted in Figure 5 of the manuscript, the $NO_2$ levels for Cluster 1 and Cluster 2 were significantly higher during the night or early morning peaks, indicating the crucial role of titration. Due to a significant reduction in NOx emissions during the static management period in 2022, the total number of days for Cluster 1 and Cluster 2 decreased from 37 days in 2020 and 40 days in 2021 to 11 days in 2022.

The common characteristic of Cluster 2 and Cluster 4 is a higher daytime net ozone production. This is attributed not only to meteorological conditions favorable for photochemical processes, characterized by high temperatures, low relative humidity, and high radiation but also to the contribution of elevated VOCs concentrations during the early morning peak, which got consumed in the ozone production process. As clearly shown in Figure 5 of the manuscript, the VOC levels during the early morning peak for Cluster 2 and Cluster 4 were significantly higher than those for Cluster 1 and Cluster 3, contributing to the higher daytime net ozone production. Due to a reduction in VOCs emissions during the static management period in 2022, the total number of days for Cluster 2 and Cluster 4 decreased from 28 days in 2020 and 25 days in 2021 to 20 days in 2022. However, the number of days for Cluster 4 increased from 4 days in 2020 and 2021 to 16 days in 2022, as the reduction in NOx emissions in 2022 led to an increase in the number of days with elevated ozone baseline levels.

Overall, the change in the number of days for the four ozone diurnal profiles is influenced by the differential reduction in precursor VOCs and NOx levels. The number of days with higher ozone concentrations for Cluster 3 and Cluster 4 significantly increased in 2022. This directly resulted in an overall elevation of ozone levels during the silent management period in 2022 compared to the same period in 2020 and 2021.

In the radical cycling OH-$RO_2$-RO-$HO_2$-OH, the OH radical initiates a series of subsequent reactions, and $RO_2$ and $HO_2$ are the primary precursors for ozone formation in the presence of NOx. The cycle is terminated through reactions with NOx under high NOx conditions and cross-reactions with ROx under low NOx conditions. We compiled the daytime average concentrations of radicals, sources of radicals, OH propagation and termination, as well as ozone production rates for Cluster 1-4 (see table R2). The concentrations and sources of radicals for Cluster 1-4 have been previously discussed in the manuscript. In summary, both the concentration and source strength of radicals were highest in Cluster 4, followed by relatively comparable levels in Cluster

2 and Cluster 3 as secondary, while Cluster 1 had the lowest levels. In the radical cycling, due to the increased consumption of VOCs, the OH propagation in Cluster 2 and Cluster 4 is significantly higher than in Cluster 1 and Cluster 3. Meanwhile, the low nitrogen oxide concentrations in Cluster 3 and Cluster 4 weaken the OH termination. This directly reflects that the ratio of OH propagation to termination in Cluster 4 was considerably higher, indicating a more efficient radical cycle. This higher efficiency allows for significant net ozone generation even in the presence of elevated baseline ozone levels.

We have included the relevant description in the manuscript; please refer to Lines 310-315 in the track-changes manuscript.

**Table R2. The ozone concentration (ppbv) characteristics, the average daytime radicals concentration (molecules cm$^{-3}$), and the average reaction rates of the main processes for Cluster 1-Cluster 4.**

|  | Cluster 1 | Cluster 2 | Cluster 3 | Cluster 4 |
|---|---|---|---|---|
| O$_3$ background | 22.4 | 16.9 | 40.6 | 33.3 |
| O$_3$ net production | 14.8 | 45.9 | 17.5 | 50.9 |
| OH concentration | $1.4\times10^6$ | $3.3\times10^6$ | $3.1\times10^6$ | $3.9\times10^6$ |
| HO$_2$ concentration | $0.6\times10^8$ | $1.5\times10^8$ | $1.9\times10^8$ | $3.5\times10^8$ |
| RO$_2$ concentration | $0.3\times10^8$ | $0.8\times10^8$ | $0.9\times10^8$ | $2.2\times10^8$ |
| P(ROx) | 0.92 | 2.09 | 1.35 | 2.23 |
| OH propagation (OH+VOCs) | 0.68 | 1.75 | 1.16 | 2.03 |
| OH termination (OH+NO$_2$) | 0.63 | 1.14 | 0.69 | 0.88 |
| propagation/termination | 1.08 | 1.54 | 1.69 | 2.29 |
| P(O$_3$) | 3.12 | 7.73 | 5.24 | 8.20 |

The dominance of radical cycling in polluted environments raises questions about its specific role in ozone formation and the potential for targeted interventions in such environments.

R: Thanks for your comments. As mentioned in the response to specific comments, an inappropriate ratio of NOx to VOCs concentrations can enhance the radical cycle, leading to substantial ozone generation within the cycling. The current mainstream perspective on ozone control emphasizes the need to establish a scientific VOCs/NOx emission reduction ratio, aiming to weaken the intensity of the radical cycling. This study suggests that reducing NO$_2$ by about 50% and VOCs by 30% in Shanghai would enhance the radical cycling, leading to an increase in ozone levels.

The spatial distribution characteristics of O$_3$ generation sensitivity exhibit both similarities and differences across different regions. Urban sites in the Beijing-Tianjin-Hebei and surrounding areas, the Yangtze River Delta, the Pearl River Delta, and other regions show consistent O$_3$ generation sensitivity, predominantly falling within the VOC control regime. (Wang et al., 2022; Wang et al., 2020a; Lyu et al., 2019; Yu et al., 2020; Wang et al., 2023). However, there are variations in the sensitivity of suburban sites. Suburban sites in the Yangtze River Delta (An et al., 2015; Xu et al., 2017; Zhao et al., 2020), Pearl River Delta (Wang et al., 2017), and Chengdu-Chongqing region (such as Chongqing) (Su et al., 2018) are mainly situated in the VOC control

regime. In contrast, suburban sites in the Beijing-Tianjin-Hebei and surrounding areas (Lu et al., 2010; Ran et al., 2011), the middle and lower reaches of the Yangtze River (such as Wuhan) (Zhu et al., 2020a), and the northwest region (such as Lanzhou) (Xue et al., 2014) are primarily located in the NOx control regime or NOx-VOC mixed control regime. The above emphasize that most areas in my country are still in a high NOx environment and VOCs play an extremely important role in the $O_3$ production process (especially $O_3$ production in urban areas). At this stage, it is still necessary to implement scientific VOCs/NOx collaborative emission reduction measures to weaken the radical cycling. Identifying and controlling key reactive VOC species in $O_3$ generation are essential for effectively mitigating $O_3$ pollution.

Minor comments:

Line 17: The statement that "the average ozone level increased by nearly 23%" lacks specific context. What does "nearly 23%" represent in terms of absolute concentrations, and how does it compare to historical averages or expected levels under similar conditions?

R: Thanks for your comments. The comparisons made here are between the static management period of 2022 and the same periods in 2020 and 2021. Therefore, the statement " the average ozone level increased by nearly 23%" refers to the average ozone concentration during the static management period of 2022 being approximately 23% higher than the same period in 2020 and 2021. We have added "Compared to 2020 and 2021" at the end of the sentence. Please refer to Line 17.

Line 39: The phrase "in those countries like China that face air pollution complex" could be better structured for clarity. Consider rephrasing to "in countries like China that face complex air pollution issues."

R: Thanks for your suggestion.  We have modified it according to your opinions. Please refer to Line 40.

Lines 40-48: The comparison with the lockdown measures in early 2020 is useful. Could the paper elaborate on how the findings from 2020 influenced the hypotheses or research methods for the 2022 study?

R: Thanks for your suggestion. This paragraph has been reorganized and revised as follows, and please refer to Lines 45-53.

"*Prior to this, China had implemented a series of nationwide lockdown measures against the occurrence and spread of the virus in early 2020. This reduction in human activity is expected to significantly reduce air pollutant emissions, as confirmed by lots of studies on lockdown in 2020 (Bao and Zhang, 2020; Huang et al., 2021; Li et al., 2021; Liu et al., 2020; Tian et al., 2021; Wang et al., 2021; Zhang et al., 2022c). Reports on the impact of the lockdown on air quality most commonly focus on measuring nitrogen dioxide ($NO_2$) and fine particulate matter ($PM_{2.5}$) (Agarwal et al., 2020; Hua et al., 2021; Chu et al., 2021). According to satellite data, tropospheric nitrogen oxides (NOx) emissions have decreased by 30-60% compared to pre-lockdown levels (Feng et al., 2020; Ding et al., 2020; Venter et al., 2020). Similarly, surface $PM_{2.5}$ levels in northern China have also decreased by approximately 35%. Meanwhile, the average $O_3$ concentration has increased 1.5-2 times (Shi and Brasseur, 2020). In Wuhan, the urban area that*

*implemented stringent measures to limit the spread of the coronavirus, concentrations of PM$_{2.5}$, NO$_2$, and ozone also exhibited similar changes (Shi and Brasseur, 2020). Most studies attribute the decrease in NO$_2$ and PM$_{2.5}$ to the suspension of transportation and industrial activities (Rana et al., 2021; Wang et al., 2020b). Huang et al. (2021) suggest that increase in O$_3$ enhances atmospheric oxidation capacity, providing favorable conditions for the formation of secondary particulate matter. Due to the lockdown taking place during winter, which is a season of high particulate matter pollution in China, the reports on the impact of the lockdown on air quality have focused more on the changes in particulate matter. The lockdown in Shanghai in 2022 was implemented in April and May during the high-ozone (O$_3$) season and lasted for a longer duration, providing an opportunity to study atmospheric pollution primarily caused by O$_3$.*"

Line 52: Ensure consistent use of punctuation, especially with regard to commas and periods. Should be "During the static management…"? Please briefly describe the definition of static management to make it clear.

R: Thanks for your suggestion. We have modified it. Please refer to Line 59. The static management is an epidemic prevention and control measure. Specifically, it means that individuals are prohibited from leaving their homes within the controlled area, residents within the controlled zone are prohibited from leaving their residential units, participation in nucleic acid sampling is restricted to in-and-out, and all citizens are required to stay indoors, avoid going out, and refrain from visiting public places unless absolutely necessary.

In Lines 32 and 33 of the manuscript, it has been explained that the Chinese government, to curb the spread of the virus, implemented control measures citywide static management in Shanghai.

Line 70: Were the experimental conditions and methodologies consistent across the different years (2020, 2021, and 2022) to ensure comparability of the data?

R: Thanks for your suggestion. Yes, the experimental conditions and methods in 2020, 2021 and 2022 are consistent, and the data are comparable.

Line 82: There seems to be a typo in the coordinates for the Yangpu environment monitoring station (31.53°N, 31.25°E). The longitude seems incorrect.

R: Thanks for your correction. The correct latitude and longitude information is 31.25°N, 121.53°E. We have made the correction. Please refer to Line 90.

Lines 107-111: The text is clear but might benefit from a brief explanation of why the stacking model is preferred over simple linear interpolation.

R: Thanks for your suggestion. Because if more than half of the hourly data is missing in a day, linear interpolation would obscure the true diurnal variation characteristics. The stacking model, however, can relatively accurately simulate the missing values.

Line 153: The methodology for classifying the 103 VOCs into six categories, including how each category specifically influences ozone formation, requires clarification. How does the classification relate to their respective roles in ozone formation? For example, the study notes a significant reduction in aromatics and a relative stability in OVOCs. What are the implications of these changes for ozone formation, considering their different sources and chemical behaviors?

R: Thanks for your comments. VOCs components are classified according to functional groups. We have modified this sentence to: "*We classified the 103 VOCs into six categories based on functional groups, including alkanes, alkenes, alkynes, aromatics, oxygenated VOCs (OVOCs), and halohydrocarbons*". Please refer to Line 162. We have added an analysis of the role of VOCs in ozone formation in the Supplement, as described in the following text and Figure R3. Please refer to Text S4 and Figure S11 in the supplement.

*"Different VOC species display a range of reactivity and diverse potentials for $O_3$ formation, which can be assessed using the maximum incremental reactivity (Carter, 2009). The calculated ozone formation potential (OFP) for each VOC species illustrates the maximum contribution of the species to the formation of ozone. (Bufalini and Dodge, 1983). The OFP for each VOC species is calculated using the following equation (Ma et al., 2019; Zhu et al., 2020b):*

$$OFP_i = MIR_i \times [VOC_i] \times \frac{M_i}{M_{ozone}} \qquad R1$$

*where $OFP_i$ (ppbv) represents the ozone formation potential of VOC species i, [VOCi] (ppbv) denotes the atmospheric concentration of VOC species i, and MIRi (g $O_3$/g VOC) is the ozone formation coefficient of VOCi in the maximum increment reactions of ozone. Mozone and Mi are the molar masses (g $mol^{-1}$) of $O_3$ and VOC species i, respectively.*

*In addition, another widely used indicator of atmospheric oxidative capacity is the OH reactivity, defined as the reaction rate coefficient multiplied by the concentration of OH reactants, depending on the abundance and composition of major pollutants. The OBM model can output the kOH for each VOCs, which reflects the reactivity of VOCs.*

*The comparison of the mean concentration of the six VOCs groups and their OFP in 2020, 2021, and 2022 is presented in Figure R3. It is obvious that the concentration of the VOC group was not proportional to its OFP. The average proportions of each VOCs component in 2020 and 2021 are 47.2%, 6.7%, 9.2%, 4.8%, 13.8% and 18.4% respectively, while the corresponding average proportions of OFP are 19.6%, 24.0%, 40.4%, 1.1%, 14.1% and 0.9%, respectively. In 2022, with minimal changes in the concentration proportions of alkenes, the OFP proportion increased by 11.3%. The proportion of aromatic hydrocarbons decreased by 4.2%, resulting in a 23.7% decrease in their OFP proportion. The proportion of OVOCs increased by 3.7%, leading to an 8.5% increase in their OFP proportion. In short, the primary contributors to OFP in 2020 and 2021 were aromatic hydrocarbons, followed by alkenes, while in 2022, the main contributors were alkenes, followed by OVOCs.*

*We combined two indicators, OFP and kOH, to identify key VOCs in photochemical processes. Figure R3b clearly shows a significant positive correlation between the OFP and kOH for each VOC after logarithmic transformation. Therefore, VOCs that rank high in both indicators are sufficient to indicate that these VOCs are key contributors in photochemical processes. In 2020 and 2021, m/p-xylene, toluene, ethylene, and propylene were the top contributors to both OFP and kOH. However, in 2022, propylene and ethylene took the lead in OFP, while propylene, isoprene, and ethylene topped the kOH."*

[Figure]

**Figure R3. (a) The mean concentration and OFP of six VOC groups and (b) Scatter plots of the average OFP and OH reactivity for individual VOCs during daytime hours between 06:00 and 18:00 for the periods of April to May in 2020, 2021, and 2022.**

Line 171: The observed delay in the peak time of VOCs in 2022 compared to 2020 and 2021 is attributed to reduced human activities. Can the study provide further evidence or analysis to support this correlation? For example, have VOC patterns changed again and peaked at around 6 am since recovering from the pandemic?

R: Thanks for your comments. The diurnal variation profiles of VOCs in June 2023 clearly illustrate a shift in the diurnal pattern of VOCs since the recovery from the pandemic, with peaks occurring around 06:00 (see Figure R4a). Additionally, a comparison between the weekday and weekend diurnal profiles in June 2023 confirms historical research, as Cai et al. (2010) suggested, that the diurnal variation of VOCs in the Shanghai area exhibits a distinct weekend effect (see Figure R4b).

We have included an analysis of this section in the Supplement, as referenced by Figure S12. Additionally, the manuscript has been updated to respond to this analysis; please refer to Lines 182-183.

[Figure]

**Figure R4. (a) The mean diurnal profiles of VOCs for the periods of April to May in 2020, 2021, and 2022, as well as June 2023. (b) The mean diurnal profiles of VOCs for weekdays and weekends in June 2023.**

Line 200: The relationship between meteorological conditions, pollution environments, and different types of ozone profiles warrants further exploration. How do these factors interact to form the distinct ozone profile clusters? Recent references (e.g., https://doi.org/10.5194/acp-18-1185-2018) also suggest that night-time $O_3$ tends to increase for some cities, would this affect the results?

R: Thanks for your comments. In the response to Specific Comments 3, the characteristics of the four ozone profiles and how precursor species $NO_2$ and VOCs influence ozone profiles were summarized. For meteorological conditions, Cluster 2 and Cluster 4 were in environments typical for photochemical processes with high temperature, low RH, and high radiation. This aligns with their corresponding ozone profile features. Cluster 3 experienced low temperature and low RH conditions, while Cluster 1 is characterized by low temperature, low RH, and low radiation, representing conditions typically unfavorable for ozone formation.

You mentioned Yan et al. (2018)'s study, where nighttime ozone concentrations showed an increasing trend in some cities. However, in this study, we compared data for the same months over three consecutive years. The variations in meteorological conditions over these three years, as detailed in the Supplement (see Figures S7-S10), were not particularly pronounced. The primary influencing factor was the reduction in pollutant emissions.

Line 221: The phrase "Reviewing previously observational results" should be "Reviewing previous observational results."

R: Thanks for your suggestion. We have modified it according to your opinions. Please refer to Line 241.

Line 248: The phrase "which has" would be more grammatically correct as "which had."

R: Thanks for your suggestion. We have modified it according to your opinions. Please refer to Line 269.

Line 253: A typo here, "sournces" should be corrected to "sources."

R: Thanks for your suggestion. We have modified it according to your opinions. Please refer to Line 276.

Line 284: The word "taked" should be corrected to "took." The sentence provides a breakdown of sources in Cluster 4.

R: Thanks for your suggestion. We have modified it according to your opinions. Please refer to Line 307.

Line 313: The phrase "It can be interfered" should be "It can be inferred."

R: Thanks for your suggestion. We have modified it according to your opinions. Please refer to Line 342.

Line 322: The phrase "which provides a valuable opportunity" is correct but could be more concise. Suggested revision: "providing a valuable opportunity."

R: Thanks for your suggestion. We have modified it according to your opinions. Please refer to

Line 323: The verb "shows" should be "show" to agree with the plural subject "observations." Remove the comma after "shows that."

R: Thanks for your suggestion.  We have modified it according to your opinions. Please refer to Line 352.

Line 329: The phrase "resulted in the overall increase in the average ozone level" could be more concise. Suggested revision: "led to an overall increase in average ozone levels."

R: Thanks for your suggestion.  We have modified it according to your opinions. Please refer to Line 358.

Line 336-337: Consider rephrasing for clarity: "The different proportions of $NO_2$ and VOC reduction during static management led to an increased OH radical propagation (OH+VOCs) to termination (OH+$NO_2$) ratio, reaching 2.10, higher than 1.03 in 2020 and 1.60 in 2021."

R: Thanks for your suggestion.  We have modified it according to your opinions. Please refer to Line 366.

Line 339: Consider rephrasing for clarity: "The important lesson from the static management 'large-scale field experiment' is that Shanghai operates in a VOCs-limited regime."

R: Thanks for your suggestion.  We have modified it according to your opinions. Please refer to Lines 368-369.

**Reference:**

Agarwal, A., Kaushik, A., Kumar, S., and Mishra, R. K.: Comparative study on air quality status in Indian and Chinese cities before and during the COVID-19 lockdown period, Air Quality, Atmosphere & Health, 13, 1167-1178, https://doi.org/10.1007/s11869-020-00881-z, 2020.

An, J., Zou, J., Wang, J., Lin, X., and Zhu, B.: Differences in ozone photochemical characteristics between the megacity Nanjing and its suburban surroundings, Yangtze River Delta, China, Environmental Science and Pollution Research, 22, 19607-19617, https://doi.org/10.1007/s11356-015-5177-0, 2015.

Bao, R. and Zhang, A.: Does lockdown reduce air pollution? Evidence from 44 cities in northern China, Science of the Total Environment, 731, 139052, https://doi.org/10.1016/j.scitotenv.2020.139052, 2020.

Bufalini, J. J. and Dodge, M. C.: Ozone-forming potential of light saturated hydrocarbons, Environmental Science & Technology, 17, 308-311, https://doi.org/10.1021/es00111a013, 1983.

Butler, T., Taraborrelli, D., Brühl, C., Fischer, H., Harder, H., Martinez, M., Williams, J., Lawrence, M., and Lelieveld, J.: Improved simulation of isoprene oxidation chemistry with the ECHAM5/MESSy chemistry-climate model: lessons from the GABRIEL airborne field campaign, Atmospheric Chemistry and Physics, 8, 4529-4546, https://doi.org/10.5194/acp-8-4529-2008, 2008.

Cai, C., Geng, F., Tie, X., Yu, Q., and An, J.: Characteristics and source apportionment of VOCs measured in Shanghai, China, Atmospheric Environment, 44, 5005-5014, https://doi.org/10.1016/j.atmosenv.2010.07.059, 2010.

Cai, C., Hogrefe, C., Katsafados, P., Kallos, G., Beauharnois, M., Schwab, J. J., Ren, X., Brune, W. H., Zhou, X., and He, Y.: Performance evaluation of an air quality forecast modeling system for a summer and winter season-Photochemical oxidants and their precursors, Atmospheric Environment, 42, 8585-8599, https://doi.org/10.1016/j.atmosenv.2008.08.029, 2008.

Carpenter, L., Fleming, Z. L., Read, K., Lee, J., Moller, S., Hopkins, J., Purvis, R., Lewis, A., Müller, K., and Heinold, B.: Seasonal characteristics of tropical marine boundary layer air measured at the Cape Verde Atmospheric Observatory, Journal of Atmospheric Chemistry, 67, 87-140, https://doi.org/10.1007/s10874-011-9206-1, 2010.

Carslaw, N., Creasey, D., Harrison, D., Heard, D., Hunter, M., Jacobs, P., Jenkin, M., Lee, J., Lewis, A., and Pilling, M.: OH and HO$_2$ radical chemistry in a forested region of north-western Greece, Atmospheric Environment, 35, 4725-4737, https://doi.org/10.1016/S1352-2310(01)00089-9, 2001.

Carslaw, N., Creasey, D., Heard, D., Jacobs, P., Lee, J., Lewis, A., McQuaid, J., Pilling, M., Bauguitte, S., and Penkett, S.: Eastern Atlantic Spring Experiment 1997 (EASE97) 2. Comparisons of model concentrations of OH, HO$_2$, and RO$_2$ with measurements, Journal of Geophysical Research: Atmospheres, 107, ACH 5-1-ACH 5-16, https://doi.org/10.1029/2001JD001568, 2002.

Carter, W. P.: Updated maximum incremental reactivity scale and hydrocarbon bin reactivities for regulatory applications, California Air Resources Board Contract, 339, 2009, 2009.

Chen, S., Ren, X., Mao, J., Chen, Z., Brune, W. H., Lefer, B., Rappenglück, B., Flynn, J., Olson, J., and Crawford, J. H.: A comparison of chemical mechanisms based on TRAMP-2006 field data, Atmospheric Environment, 44, 4116-4125, https://doi.org/10.1016/j.atmosenv.2009.05.027, 2010.

Chu, B., Zhang, S., Liu, J., Ma, Q., and He, H.: Significant concurrent decrease in PM$_{2.5}$ and NO$_2$ concentrations in China during COVID-19 epidemic, Journal of Environmental Sciences, 99, 346-353, https://doi.org/10.1016/j.jes.2020.06.031, 2021.

Creasey, D., Heard, D., and Lee, J.: OH and HO$_2$ measurements in a forested region of north-western Greece, Atmospheric Environment, 35, 4713-4724, https://doi.org/10.1016/S1352-2310(01)00090-5, 2001.

Creasey, D., Heard, D., and Lee, J.: Eastern Atlantic Spring Experiment 1997 (EASE97) 1. Measurements of OH and HO$_2$ concentrations at Mace Head, Ireland, Journal of Geophysical Research: Atmospheres, 107, ACH 3-1-ACH 3-15, https://doi.org/10.1029/2001JD000892, 2002.

Creasey, D., Evans, G., Heard, D., and Lee, J.: Measurements of OH and HO$_2$ concentrations in the Southern Ocean marine boundary layer, Journal of Geophysical Research: Atmospheres, 108, https://doi.org/10.1029/2002JD003206, 2003.

Ding, J., van der A, R. J., Eskes, H., Mijling, B., Stavrakou, T., Van Geffen, J., and Veefkind, J.: NOx emissions reduction and rebound in China due to the COVID-19 crisis, Geophysical Research Letters, 47, e2020GL089912, https://doi.org/10.1029/2020GL089912, 2020.

Dolgorouky, C., Gros, V., Sarda-Esteve, R., Sinha, V., Williams, J., Marchand, N., Sauvage, S., Poulain, L., Sciare, J., and Bonsang, B.: Total OH reactivity measurements in Paris during the 2010 MEGAPOLI winter campaign, Atmospheric Chemistry and Physics, 12, 9593-9612, https://doi.org/10.5194/acp-12-9593-2012, 2012.

Dusanter, S., Vimal, D., Stevens, P., Volkamer, R., and Molina, L.: Measurements of OH and HO$_2$ concentrations during the MCMA-2006 field campaign-Part 1: Deployment of the Indiana University

laser-induced fluorescence instrument, Atmospheric Chemistry and Physics, 9, 1665-1685, https://doi.org/10.5194/acp-9-1665-2009, 2009a.

Dusanter, S., Vimal, D., Stevens, P., Volkamer, R., Molina, L., Baker, A., Meinardi, S., Blake, D., Sheehy, P., and Merten, A.: Measurements of OH and $HO_2$ concentrations during the MCMA-2006 field campaign-Part 2: Model comparison and radical budget, Atmospheric Chemistry and Physics, 9, 6655-6675, https://doi.org/10.5194/acp-9-6655-2009, 2009b.

Emmerson, K. and Carslaw, N.: Night-time radical chemistry during the TORCH campaign, Atmospheric Environment, 43, 3220-3226, https://doi.org/10.1016/j.atmosenv.2009.03.042, 2009.

Emmerson, K., Carslaw, N., and Pilling, M.: Urban atmospheric chemistry during the PUMA campaign 2: Radical budgets for OH, $HO_2$ and $RO_2$, Journal of atmospheric chemistry, 52, 165-183, https://doi.org/10.1007/s10874-005-1323-2, 2005a.

Emmerson, K., Carslaw, N., Carpenter, L., Heard, D., Lee, J., and Pilling, M.: Urban atmospheric chemistry during the PUMA campaign 1: Comparison of modelled OH and $HO_2$ concentrations with measurements, Journal of Atmospheric Chemistry, 52, 143-164, https://doi.org/10.1007/s10874-005-1322-3, 2005b.

Emmerson, K. M., Carslaw, N., Carslaw, D., Lee, J. D., McFiggans, G., Bloss, W. J., Gravestock, T., Heard, D. E., Hopkins, J., and Ingham, T.: Free radical modelling studies during the UK TORCH Campaign in Summer 2003, Atmospheric Chemistry and Physics, 7, 167-181, https://doi.org/10.5194/acp-7-167-2007, 2007.

Faloona, I., Tan, D., Brune, W., Hurst, J., Barket Jr, D., Couch, T. L., Shepson, P., Apel, E., Riemer, D., and Thornberry, T.: Nighttime observations of anomalously high levels of hydroxyl radicals above a deciduous forest canopy, Journal of Geophysical Research: Atmospheres, 106, 24315-24333, https://doi.org/10.1029/2000JD900691, 2001.

Feng, S., Jiang, F., Wang, H., Wang, H., Ju, W., Shen, Y., Zheng, Y., Wu, Z., and Ding, A.: NOx emission changes over China during the COVID-19 epidemic inferred from surface NO2 observations, Geophysical Research Letters, 47, e2020GL090080, https://doi.org/10.1029/2020GL090080, 2020.

Fuchs, H., Tan, Z., Lu, K., Bohn, B., Broch, S., Brown, S. S., Dong, H., Gomm, S., Häseler, R., and He, L.: OH reactivity at a rural site (Wangdu) in the North China Plain: contributions from OH reactants and experimental OH budget, Atmospheric Chemistry and Physics, 17, 645-661, https://doi.org/10.5194/acp-17-645-2017, 2017.

Harrison, R., Yin, J., Tilling, R., Cai, X., Seakins, P., Hopkins, J., Lansley, D., Lewis, A., Hunter, M., and Heard, D.: Measurement and modelling of air pollution and atmospheric chemistry in the UK West Midlands conurbation: Overview of the PUMA Consortium project, Science of the Total Environment, 360, 5-25, https://doi.org/10.1016/j.scitotenv.2005.08.053, 2006.

Heard, D., Carpenter, L., Creasey, D., Hopkins, J., Lee, J., Lewis, A., Pilling, M., Seakins, P., Carslaw, N., and Emmerson, K.: High levels of the hydroxyl radical in the winter urban troposphere, Geophysical research letters, 31, https://doi.org/10.1029/2004GL020544, 2004.

Hens, K., Novelli, A., Martinez, M., Auld, J., Axinte, R., Bohn, B., Fischer, H., Keronen, P., Kubistin, D., and Nölscher, A.: Observation and modelling of HO x radicals in a boreal forest, Atmospheric Chemistry and Physics, 14, 8723-8747, https://doi.org/10.5194/acp-14-8723-2014, 2014.

Hua, J., Zhang, Y., de Foy, B., Shang, J., Schauer, J. J., Mei, X., Sulaymon, I. D., and Han, T.: Quantitative estimation of meteorological impacts and the COVID-19 lockdown reductions on $NO_2$ and $PM_{2.5}$ over the Beijing area using Generalized Additive Models (GAM), Journal of Environmental Management, 291, 112676, https://doi.org/10.1016/j.jenvman.2021.112676, 2021.

Huang, X., Ding, A., Gao, J., Zheng, B., Zhou, D., Qi, X., Tang, R., Wang, J., Ren, C., and Nie, W.: Enhanced secondary pollution offset reduction of primary emissions during COVID-19 lockdown in China, National Science Review, 8, nwaa137, https://doi.org/10.1093/nsr/nwaa137, 2021.

Kanaya, Y. and Akimoto, H.: Direct measurements of HOx radicals in the marine boundary layer: Testing the current tropospheric chemistry mechanism, The Chemical Record, 2, 199-211, https://doi.org/10.1002/tcr.10019, 2002.

Kanaya, Y., Matsumoto, J., and Akimoto, H.: Photochemical ozone production at a subtropical island of Okinawa, Japan: Implications from simultaneous observations of $HO_2$ radical and NOx, Journal of Geophysical Research: Atmospheres, 107, ACH 2-1-ACH 2-14, https://doi.org/10.1029/2001JD000858, 2002a.

Kanaya, Y., Matsumoto, J., Kato, S., and Akimoto, H.: Behavior of OH and $HO_2$ radicals during the Observations at a Remote Island of Okinawa (ORION99) field campaign: 2. Comparison between observations and calculations, Journal of Geophysical Research: Atmospheres, 106, 24209-24223, https://doi.org/10.1029/2000JD000179, 2001a.

Kanaya, Y., Sadanaga, Y., Nakamura, K., and Akimoto, H.: Behavior of OH and $HO_2$ radicals during the Observations at a Remote Island of Okinawa (ORION99) field campaign: 1. Observation using a

laser-induced fluorescence instrument, Journal of Geophysical Research: Atmospheres, 106, 24197-24208, https://doi.org/10.1029/2000JD000178, 2001b.

Kanaya, Y., Nakamura, K., Kato, S., Matsumoto, J., Tanimoto, H., and Akimoto, H.: Nighttime variations in $HO_2$ radical mixing ratios at Rishiri Island observed with elevated monoterpene mixing ratios, Atmospheric Environment, 36, 4929-4940, https://doi.org/10.1016/S1352-2310(02)00401-6, 2002b.

Kanaya, Y., Sadanaga, Y., Matsumoto, J., Sharma, U. K., Hirokawa, J., Kajii, Y., and Akimoto, H.: Daytime $HO_2$ concentrations at Oki Island, Japan, in summer 1998: Comparison between measurement and theory, Journal of Geophysical Research: Atmospheres, 105, 24205-24222, https://doi.org/10.1029/2000JD900308, 2000.

Kanaya, Y., Fukuda, M., Akimoto, H., Takegawa, N., Komazaki, Y., Yokouchi, Y., Koike, M., and Kondo, Y.: Urban photochemistry in central Tokyo: 2. Rates and regimes of oxidant ($O_3$+ $NO_2$) production, Journal of Geophysical Research: Atmospheres, 113, https://doi.org/10.1029/2007JD008671, 2008.

Kanaya, Y., Yokouchi, Y., Matsumoto, J., Nakamura, K., Kato, S., Tanimoto, H., Furutani, H., Toyota, K., and Akimoto, H.: Implications of iodine chemistry for daytime $HO_2$ levels at Rishiri Island, Geophysical research letters, 29, 45-41-45-44, https://doi.org/10.1029/2001GL014061, 2002c.

Kanaya, Y., Cao, R., Akimoto, H., Fukuda, M., Komazaki, Y., Yokouchi, Y., Koike, M., Tanimoto, H., Takegawa, N., and Kondo, Y.: Urban photochemistry in central Tokyo: 1. Observed and modeled OH and $HO_2$ radical concentrations during the winter and summer of 2004, Journal of Geophysical Research: Atmospheres, 112, https://doi.org/10.1029/2007JD008670, 2007a.

Kanaya, Y., Cao, R., Kato, S., Miyakawa, Y., Kajii, Y., Tanimoto, H., Yokouchi, Y., Mochida, M., Kawamura, K., and Akimoto, H.: Chemistry of OH and $HO_2$ radicals observed at Rishiri Island, Japan, in September 2003: Missing daytime sink of $HO_2$ and positive nighttime correlations with monoterpenes, Journal of Geophysical Research: Atmospheres, 112, https://doi.org/10.1029/2006JD007987, 2007b.

Kanaya, Y., Hofzumahaus, A., Dorn, H.-P., Brauers, T., Fuchs, H., Holland, F., Rohrer, F., Bohn, B., Tillmann, R., and Wegener, R.: Comparisons of observed and modeled OH and $HO_2$ concentrations during the ambient measurement period of the HOx Comp field campaign, Atmospheric Chemistry and Physics, 12, 2567-2585, https://doi.org/10.5194/acp-12-2567-2012, 2012.

Kovacs, T., Brune, W., Harder, H., Martinez, M., Simpas, J., Frost, G., Williams, E., Jobson, T., Stroud, C., and Young, V.: Direct measurements of urban OH reactivity during Nashville SOS in summer 1999, Journal of Environmental Monitoring, 5, 68-74, https://doi.org/10.1039/B204339D, 2003.

Kubistin, D., Harder, H., Martinez, M., Rudolf, M., Sander, R., Bozem, H., Eerdekens, G., Fischer, H., Gurk, C., and Klüpfel, T.: Hydroxyl radicals in the tropical troposphere over the Suriname rainforest: comparison of measurements with the box model MECCA, Atmospheric Chemistry and Physics, 10, 9705-9728, https://doi.org/10.5194/acp-10-9705-2010, 2010.

Lee, J., McFiggans, G., Allan, J., Baker, A., Ball, S., Benton, A., Carpenter, L., Commane, R., Finley, B., and Evans, M.: Reactive halogens in the marine boundary layer (RHaMBLe): the tropical North Atlantic experiments, Atmospheric Chemistry and Physics, 10, 1031-1055, https://doi.org/10.5194/acp-10-1031-2010, 2010.

Lee, J. D., Lewis, A. C., Monks, P. S., Jacob, M., Hamilton, J. F., Hopkins, J. R., Watson, N. M., Saxton, J. E., Ennis, C., and Carpenter, L. J.: Ozone photochemistry and elevated isoprene during the UK heatwave of August 2003, Atmospheric Environment, 40, 7598-7613, https://doi.org/10.1016/j.atmosenv.2006.06.057, 2006.

Lelieveld, J. a., Butler, T., Crowley, J., Dillon, T., Fischer, H., Ganzeveld, L., Harder, H., Lawrence, M., Martinez, M., and Taraborrelli, D.: Atmospheric oxidation capacity sustained by a tropical forest, Nature, 452, 737-740, https://doi.org/10.1038/nature06870, 2008.

Li, R., Zhao, Y., Fu, H., Chen, J., Peng, M., and Wang, C.: Substantial changes in gaseous pollutants and chemical compositions in fine particles in the North China Plain during the COVID-19 lockdown period: anthropogenic vs. meteorological influences, Atmospheric Chemistry and Physics, 21, 8677-8692, https://doi.org/10.5194/acp-21-8677-2021, 2021.

Liu, T., Wang, X., Hu, J., Wang, Q., An, J., Gong, K., Sun, J., Li, L., Qin, M., and Li, J.: Driving forces of changes in air quality during the COVID-19 lockdown period in the Yangtze River Delta Region, China, Environmental Science & Technology Letters, 7, 779-786, https://doi.org/10.1021/acs.estlett.0c00511, 2020.

LU, K.-D. and ZHANG, Y.-H.: Observations of HOx Radical in Field Studies and the Analysisi of Its Chemical Mechanism, Progress in Chemistry, 22, 500, 2010.

Lu, K., Guo, S., Tan, Z., Wang, H., Shang, D., Liu, Y., Li, X., Wu, Z., Hu, M., and Zhang, Y.: Exploring atmospheric free-radical chemistry in China: the self-cleansing capacity and the formation of secondary

air pollution, National Science Review, 6, 579-594, https://doi.org/10.1093/nsr/nwy073, 2019.

Lu, K., Zhang, Y., Su, H., Brauers, T., Chou, C. C., Hofzumahaus, A., Liu, S. C., Kita, K., Kondo, Y., and Shao, M.: Oxidant ($O_3$+ $NO_2$) production processes and formation regimes in Beijing, Journal of Geophysical Research: Atmospheres, 115, https://doi.org/10.1029/2009JD012714, 2010.

Lyu, X., Wang, N., Guo, H., Xue, L., Jiang, F., Zeren, Y., Cheng, H., Cai, Z., Han, L., and Zhou, Y.: Causes of a continuous summertime $O_3$ pollution event in Jinan, a central city in the North China Plain, Atmospheric Chemistry and Physics, 19, 3025-3042, https://doi.org/10.5194/acp-19-3025-2019, 2019.

Ma, Z., Liu, C., Zhang, C., Liu, P., Ye, C., Xue, C., Zhao, D., Sun, J., Du, Y., and Chai, F.: The levels, sources and reactivity of volatile organic compounds in a typical urban area of Northeast China, Journal of Environmental Sciences, 79, 121-134, https://doi.org/10.1016/j.jes.2018.11.015, 2019.

Mao, J., Ren, X., Chen, S., Brune, W. H., Chen, Z., Martinez, M., Harder, H., Lefer, B., Rappenglück, B., and Flynn, J.: Atmospheric oxidation capacity in the summer of Houston 2006: Comparison with summer measurements in other metropolitan studies, Atmospheric Environment, 44, 4107-4115, https://doi.org/10.1016/j.atmosenv.2009.01.013, 2010.

Martinez, M., Harder, H., Kovacs, T., Simpas, J., Bassis, J., Lesher, R., Brune, W., Frost, G., Williams, E., and Stroud, C.: OH and $HO_2$ concentrations, sources, and loss rates during the Southern Oxidants Study in Nashville, Tennessee, summer 1999, Journal of Geophysical Research: Atmospheres, 108, https://doi.org/10.1029/2003JD003551, 2003.

Martinez, M., Harder, H., Kubistin, D., Rudolf, M., Bozem, H., Eerdekens, G., Fischer, H., Klüpfel, T., Gurk, C., and Königstedt, R.: Hydroxyl radicals in the tropical troposphere over the Suriname rainforest: airborne measurements, Atmospheric Chemistry and Physics, 10, 3759-3773, https://doi.org/10.5194/acp-10-3759-2010, 2010.

Michoud, V., Kukui, A., Camredon, M., Colomb, A., Borbon, A., Miet, K., Aumont, B., Beekmann, M., Durand-Jolibois, R., and Perrier, S.: Radical budget analysis in a suburban European site during the MEGAPOLI summer field campaign, Atmospheric Chemistry and Physics, 12, 11951-11974, https://doi.org/10.5194/acp-12-11951-2012, 2012.

Nölscher, A., Williams, J., Sinha, V., Custer, T., Song, W., Johnson, A., Axinte, R., Bozem, H., Fischer, H., and Pouvesle, N.: Summertime total OH reactivity measurements from boreal forest during HUMPPA-COPEC 2010, Atmospheric chemistry and Physics, 12, 8257-8270, https://doi.org/10.5194/acp-12-8257-2012, 2012.

Pugh, T., MacKenzie, A., Hewitt, C., Langford, B., Edwards, P., Furneaux, K., Heard, D., Hopkins, J., Jones, C., and Karunaharan, A.: Simulating atmospheric composition over a South-East Asian tropical rainforest: performance of a chemistry box model, Atmospheric Chemistry and Physics, 10, 279-298, https://doi.org/10.5194/acp-10-279-2010, 2010.

Qi, B., Kanaya, Y., Takami, A., Hatakeyama, S., Kato, S., Sadanaga, Y., Tanimoto, H., and Kajii, Y.: Diurnal peroxy radical chemistry at a remote coastal site over the sea of Japan, Journal of Geophysical Research: Atmospheres, 112, https://doi.org/10.1029/2006JD008236, 2007.

Ran, L., Zhao, C., Xu, W., Lu, X., Han, M., Lin, W., Yan, P., Xu, X., Deng, Z., and Ma, N.: VOC reactivity and its effect on ozone production during the HaChi summer campaign, Atmospheric Chemistry and Physics, 11, 4657-4667, https://doi.org/10.5194/acp-11-4657-2011, 2011.

Rana, R. H., Keramat, S. A., and Gow, J.: A systematic literature review of the impact of COVID-19 lockdowns on air quality in China, Aerosol and Air Quality Research, 21, 200614, https://doi.org/10.4209/aaqr.200614, 2021.

Ren, X., Harder, H., Martinez, M., Lesher, R. L., Oliger, A., Shirley, T., Adams, J., Simpas, J. B., and Brune, W. H.: HOx concentrations and OH reactivity observations in New York City during PMTACS-NY2001, Atmospheric Environment, 37, 3627-3637, https://doi.org/10.1016/S1352-2310(03)00460-6, 2003a.

Ren, X., Brune, W. H., Mao, J., Mitchell, M. J., Lesher, R. L., Simpas, J. B., Metcalf, A. R., Schwab, J. J., Cai, C., and Li, Y.: Behavior of OH and $HO_2$ in the winter atmosphere in New York City, Atmospheric Environment, 40, 252-263, https://doi.org/10.1016/j.atmosenv.2005.11.073, 2006.

Ren, X., Harder, H., Martinez, M., Lesher, R. L., Oliger, A., Simpas, J. B., Brune, W. H., Schwab, J. J., Demerjian, K. L., and He, Y.: OH and $HO_2$ chemistry in the urban atmosphere of New York City, Atmospheric Environment, 37, 3639-3651, https://doi.org/10.1016/S1352-2310(03)00459-X, 2003b.

Sheehy, P., Volkamer, R., Molina, L. T., and Molina, M. J.: Oxidative capacity of the Mexico City atmosphere-Part 2: A RO x radical cycling perspective, Atmospheric Chemistry and Physics, 10, 6993-7008, https://doi.org/10.5194/acp-10-6993-2010, 2010.

Shi, X. and Brasseur, G. P.: The response in air quality to the reduction of Chinese economic activities during the COVID-19 outbreak, Geophysical Research Letters, 47, e2020GL088070, https://doi.org/10.1029/2020GL088070, 2020.

Shirley, T., Brune, W., Ren, X., Mao, J., Lesher, R., Cardenas, B., Volkamer, R., Molina, L., Molina, M.

J., and Lamb, B.: Atmospheric oxidation in the Mexico city metropolitan area (MCMA) during April 2003, Atmospheric Chemistry and Physics, 6, 2753-2765, https://doi.org/10.5194/acp-6-2753-2006, 2006.

Sommariva, R., Haggerstone, A.-L., Carpenter, L., Carslaw, N., Creasey, D., Heard, D., Lee, J., Lewis, A., Pilling, M., and Zádor, J.: OH and $HO_2$ chemistry in clean marine air during SOAPEX-2, Atmospheric Chemistry and Physics, 4, 839-856, https://doi.org/10.5194/acp-4-839-2004, 2004.

Stone, D., Evans, M. J., Edwards, P., Commane, R., Ingham, T., Rickard, A., Brookes, D., Hopkins, J., Leigh, R., and Lewis, A.: Isoprene oxidation mechanisms: measurements and modelling of OH and $HO_2$ over a South-East Asian tropical rainforest during the OP3 field campaign, Atmospheric Chemistry and Physics, 11, 6749-6771, https://doi.org/10.5194/acp-11-6749-2011, 2011.

Su, R., Lu, K., Yu, J., Tan, Z., Jiang, M., Li, J., Xie, S., Wu, Y., Zeng, L., and Zhai, C.: Exploration of the formation mechanism and source attribution of ambient ozone in Chongqing with an observation-based model, Science China Earth Sciences, 61, 23-32, https://doi.org/10.1007/s11430-017-9104-9, 2018.

Tan, D., Faloona, I., Simpas, J., Brune, W., Shepson, P., Couch, T., Sumner, A., Carroll, M., Thornberry, T., and Apel, E.: HOx budgets in a deciduous forest: Results from the PROPHET summer 1998 campaign, Journal of Geophysical Research: Atmospheres, 106, 24407-24427, https://doi.org/10.1029/2001JD900016, 2001.

Tan, Z., Fuchs, H., Lu, K., Hofzumahaus, A., Bohn, B., Broch, S., Dong, H., Gomm, S., Häseler, R., and He, L.: Radical chemistry at a rural site (Wangdu) in the North China Plain: observation and model calculations of OH, $HO_2$ and $RO_2$ radicals, Atmospheric Chemistry and Physics, 17, 663-690, https://doi.org/10.5194/acp-17-663-2017, 2017.

Tan, Z., Lu, K., Hofzumahaus, A., Fuchs, H., Bohn, B., Holland, F., Liu, Y., Rohrer, F., Shao, M., and Sun, K.: Experimental budgets of OH, $HO_2$, and $RO_2$ radicals and implications for ozone formation in the Pearl River Delta in China 2014, Atmospheric chemistry and physics, 19, 7129-7150, https://doi.org/10.5194/acp-19-7129-2019, 2019.

Tan, Z., Rohrer, F., Lu, K., Ma, X., Bohn, B., Broch, S., Dong, H., Fuchs, H., Gkatzelis, G. I., and Hofzumahaus, A.: Wintertime photochemistry in Beijing: observations of ROx radical concentrations in the North China Plain during the BEST-ONE campaign, Atmospheric Chemistry and Physics, 18, 12391-12411, https://doi.org/10.5194/acp-18-12391-2018, 2018.

Thornton, J., Wooldridge, P., Cohen, R., Martinez, M., Harder, H., Brune, W., Williams, E., Roberts, J., Fehsenfeld, F., and Hall, S.: Ozone production rates as a function of NOx abundances and HOx production rates in the Nashville urban plume, Journal of Geophysical Research: Atmospheres, 107, ACH 7-1-ACH 7-17, https://doi.org/10.1029/2001JD000932, 2002.

Tian, J., Wang, Q., Zhang, Y., Yan, M., Liu, H., Zhang, N., Ran, W., and Cao, J.: Impacts of primary emissions and secondary aerosol formation on air pollution in an urban area of China during the COVID-19 lockdown, Environment International, 150, 106426, https://doi.org/10.1016/j.envint.2021.106426, 2021.

Vaughan, S., Ingham, T., Whalley, L., Stone, D., Evans, M., Read, K., Lee, J., Moller, S., Carpenter, L., and Lewis, A.: Seasonal observations of OH and $HO_2$ in the remote tropical marine boundary layer, Atmospheric Chemistry and Physics, 12, 2149-2172, https://doi.org/10.5194/acp-12-2149-2012, 2012.

Venter, Z. S., Aunan, K., Chowdhury, S., and Lelieveld, J.: COVID-19 lockdowns cause global air pollution declines, Proceedings of the National Academy of Sciences, 117, 18984-18990, https://doi.org/10.1073/pnas.200685311, 2020.

Volkamer, R., Sheehy, P., Molina, L. T., and Molina, M. J.: Oxidative capacity of the Mexico City atmosphere-Part 1: A radical source perspective, Atmospheric Chemistry and Physics, 10, 6969-6991, https://doi.org/10.5194/acp-10-6969-2010, 2010.

Wang, M., Chen, W., Zhang, L., Qin, W., Zhang, Y., Zhang, X., and Xie, X.: Ozone pollution characteristics and sensitivity analysis using an observation-based model in Nanjing, Yangtze River Delta Region of China, Journal of Environmental Sciences, 93, 13-22, https://doi.org/10.1016/j.jes.2020.02.027, 2020a.

Wang, N., Xu, J., Pei, C., Tang, R., Zhou, D., Chen, Y., Li, M., Deng, X., Deng, T., and Huang, X.: Air quality during COVID-19 lockdown in the Yangtze River Delta and the Pearl River Delta: Two different responsive mechanisms to emission reductions in China, Environmental Science & Technology, 55, 5721-5730, https://doi.org/10.1021/acs.est.0c08383, 2021.

Wang, P., Chen, K., Zhu, S., Wang, P., and Zhang, H.: Severe air pollution events not avoided by reduced anthropogenic activities during COVID-19 outbreak, Resources, Conservation and Recycling, 158, 104814, https://doi.org/10.1016/j.resconrec.2020.104814, 2020b.

Wang, X., Yin, S., Zhang, R., Yuan, M., and Ying, Q.: Assessment of summertime $O_3$ formation and the $O_3$-NOx-VOC sensitivity in Zhengzhou, China using an observation-based model, Science of the Total

Environment, 813, 152449, https://doi.org/10.1016/j.scitotenv.2021.152449, 2022.

Wang, Y., Wang, H., Guo, H., Lyu, X., Cheng, H., Ling, Z., Louie, P. K., Simpson, I. J., Meinardi, S., and Blake, D. R.: Long-term $O_3$-precursor relationships in Hong Kong: field observation and model simulation, Atmospheric Chemistry and Physics, 17, 10919-10935, https://doi.org/10.5194/acp-17-10919-2017, 2017.

Wang, Z., Zhang, P., Pan, L., Qian, Y., Li, Z., Li, X., Guo, C., Zhu, X., Xie, Y., and Wei, Y.: Ambient Volatile Organic Compound Characterization, Source Apportionment, and Risk Assessment in Three Megacities of China in 2019, Toxics, 11, 651, https://doi.org/10.3390/toxics11080651, 2023.

Whalley, L., Edwards, P., Furneaux, K., Goddard, A., Ingham, T., Evans, M. J., Stone, D., Hopkins, J., Jones, C. E., and Karunaharan, A.: Quantifying the magnitude of a missing hydroxyl radical source in a tropical rainforest, Atmospheric Chemistry and Physics, 11, 7223-7233, https://doi.org/10.5194/acp-11-7223-2011, 2011.

Whalley, L., Furneaux, K., Goddard, A., Lee, J., Mahajan, A., Oetjen, H., Read, K., Kaaden, N., Carpenter, L., and Lewis, A.: The chemistry of OH and $HO_2$ radicals in the boundary layer over the tropical Atlantic Ocean, Atmospheric Chemistry and Physics, 10, 1555-1576, https://doi.org/10.5194/acp-10-1555-2010, 2010.

Xu, Z., Huang, X., Nie, W., Chi, X., Xu, Z., Zheng, L., Sun, P., and Ding, A.: Influence of synoptic condition and holiday effects on VOCs and ozone production in the Yangtze River Delta region, China, Atmospheric Environment, 168, 112-124, https://doi.org/10.1016/j.atmosenv.2017.08.035, 2017.

Xue, L., Wang, T., Gao, J., Ding, A., Zhou, X., Blake, D., Wang, X., Saunders, S., Fan, S., and Zuo, H.: Ground-level ozone in four Chinese cities: precursors, regional transport and heterogeneous processes, Atmospheric Chemistry and Physics, 14, 13175-13188, 2014.

Yan, Y., Lin, J., and He, C.: Ozone trends over the United States at different times of day, Atmospheric Chemistry and Physics, 18, 1185-1202, https://doi.org/10.5194/acp-18-1185-2018, 2018.

Yang, X., Lu, K., Ma, X., Gao, Y., Tan, Z., Wang, H., Chen, X., Li, X., Huang, X., and He, L.: Radical chemistry in the Pearl River Delta: observations and modeling of OH and $HO_2$ radicals in Shenzhen in 2018, Atmospheric Chemistry and Physics, 22, 12525-12542, https://doi.org/10.5194/acp-22-12525-2022, 2022.

Yang, X., Lu, K., Ma, X., Liu, Y., Wang, H., Hu, R., Li, X., Lou, S., Chen, S., and Dong, H.: Observations and modeling of OH and $HO_2$ radicals in Chengdu, China in summer 2019, Science of The Total Environment, 772, 144829, https://doi.org/10.1016/j.scitotenv.2020.144829, 2021.

Yu, D., Tan, Z., Lu, K., Ma, X., Li, X., Chen, S., Zhu, B., Lin, L., Li, Y., and Qiu, P.: An explicit study of local ozone budget and NOx-VOCs sensitivity in Shenzhen China, Atmospheric Environment, 224, 117304, https://doi.org/10.1016/j.atmosenv.2020.117304, 2020.

Zhang, G., Hu, R., Xie, P., Lou, S., Wang, F., Wang, Y., Qin, M., Li, X., Liu, X., and Wang, Y.: Observation and simulation of HOx radicals in an urban area in Shanghai, China, Science of The Total Environment, 810, 152275, https://doi.org/10.1016/j.scitotenv.2021.152275, 2022a.

Zhang, G., Hu, R., Xie, P., Lu, K., Lou, S., Liu, X., Li, X., Wang, F., Wang, Y., and Yang, X.: Intercomparison of OH radical measurement in a complex atmosphere in Chengdu, China, Science of the Total Environment, 838, 155924, https://doi.org/10.1016/j.scitotenv.2022.155924, 2022b.

Zhang, K., Liu, Z., Zhang, X., Li, Q., Jensen, A., Tan, W., Huang, L., Wang, Y., de Gouw, J., and Li, L.: Insights into the significant increase in ozone during COVID-19 in a typical urban city of China, Atmospheric Chemistry and Physics, 22, 4853-4866, https://doi.org/10.5194/acp-22-4853-2022, 2022c.

Zhao, Y., Chen, L., Li, K., Han, L., Zhang, X., Wu, X., Gao, X., Azzi, M., and Cen, K.: Atmospheric ozone chemistry and control strategies in Hangzhou, China: Application of a 0-D box model, Atmospheric Research, 246, 105109, https://doi.org/10.1016/j.atmosres.2020.105109, 2020.

Zhu, J., Cheng, H., Peng, J., Zeng, P., Wang, Z., Lyu, X., and Guo, H.: $O_3$ photochemistry on $O_3$ episode days and non-$O_3$ episode days in Wuhan, Central China, Atmospheric Environment, 223, 117236, https://doi.org/10.1016/j.atmosenv.2019.117236, 2020a.

Zhu, J., Wang, S., Wang, H., Jing, S., Lou, S., Saiz-Lopez, A., and Zhou, B.: Observationally constrained modeling of atmospheric oxidation capacity and photochemical reactivity in Shanghai, China, Atmospheric Chemistry and Physics, 20, 1217-1232, https://doi.org/10.5194/acp-20-1217-2020, 2020b.

---

## Author Comment (AC2)

The authors present a comprehensive analysis from the perspective in both observations and model simulation to uncover the reason for the high-level Ozone concentration during Shanghai's static management. And by comparing the similar experiments conducted during the same period in 2020 and 2021, they find that the average concentration of ozone was nearly 23% higher during the static management compared to 2020 and 2021, despite the concentrations of VOCs and NOx decreased approximately 30% and 50%. With cluster analysis of diurnal patterns of ozone concentration, they conclude that the increasing days with high ozone levels leads to an overall high average concentration of ozone during the static management. And with a model to simulate the chemical process, they find that the higher $VOCs/NO_2$ ratio during the static management strengthens the radical cycle and which leads to an active photochemical process.

First, thank you for reviewing my manuscript. We found some issues with the photolysis data and have rerun the OBM model. We've also rechecked the results in Section 3.3. The new findings don't change the original conclusions much. My replies to your comments are based on these new results.

There are the comments to improve the manuscript: Reference:

Some of the references are missed in the main context, for example:

(1) line 33-35, (2) line 158-159, (3) line 231-234

R: Thanks for your correction. We have inserted relevant reference information, please refer to Line 34, Line 167, and Line 252.

Typo, format and description:

(1) line 52: during (should be During)

R: Thanks for your correction. We have modified it, please refer to Line 59.

(2) may be simply the units for the whole manuscript, for example, in line 129: "36.8±24.1 ug m$^{-3}$, 30.0±23.1 ug m$^{-3}$, and 21.8±14.0 ug m$^{-3}$" can be written as "36.8±24.1, 30.0±23.1, and 21.8±14.0 ug m$^{-3}$", and this information is related to Figure 1d, it should be noticed.

R: Thanks for your correction. We have modified it, please refer to Line 136.

(3) some of the arrows in Figure 4d were submerged and not able to see

R: Thanks for your correction. To be more reader-friendly, we've enlarged the arrows and changed their colors, as shown in Figure R1.

[Figure]

**Figure R1. (a) Comparison of the mean diurnal profiles of the four types of O₃ after clustering. Colored areas denote 95% confidence intervals; (b) The proportions of the four clusters in 2020, 2021 and 2022. (c) Comparison of the O₃ levels of the four clusters in 2020, 2021 and 2022. The top and bottom of the vertical line for each box correspond to the 95th and 5th percentiles, respectively. The dots represent the averages, and the top, middle, and bottom lines of the box mark the 75th, 50th, and 25th percentiles, respectively; (d) Comparison of the average ozone concentrations in 2020, 2021 and 2022 for different ratios of the four clusters.**

(4) the content in line 201-203 is corresponding to Figure 5, it should be noticed

R: Thanks for your correction. We have modified it, please refer to Line 217.

(5) nothing was written for the PM₂.₅ profiles in Figure 5d

R: Thank you for your reminder. The daily variations of PM₂.₅ under four clusters exhibit similar patterns to those of VOCs. We have added the "*The diurnal profiles of PM2.5 under four clusters exhibit similar patterns to those of VOCs, with Cluster 2 and Cluster 4 exhibiting a distinct morning peak.*" to the manuscript. Please refer to Lines 225-226.

(6) line 219: the unit of OH was missed, and the corresponding years should be specified

R: Thanks for your correction. We have modified it, please refer to Line 237.

(7) line 247: the (average) rate of

R: Thanks for your correction. We have modified it, please refer to Line 268.

(8) line 248: "which has an average rate of around 0.02 ppbv h-1", does this average rate is for the three years? It should be noticed.

R: Thanks for your comment. Yes, this average rate is for the three years. We have noticed it, please refer to Line 270.

(9) line 254: RO₂ (should be RO₂)

R: Thanks for your correction. We have modified it, please refer to Line 275.

(10) line 274: cm³ (should be cm⁻³)

R: Thanks for your correction. We have modified it, please refer to Line 295.

(11) line space and post paragraph space for the title of Figure 7.

R: Thanks for your correction. We have modified it, please refer to Line 316.

(12) line 313: interfered (should be inferred)

R: Thanks for your correction. We have modified it, please refer to Line 342.

(13) line 323: that , there (should be that, there)

R: Thanks for your correction. We have modified it, please refer to Line 352.

(14) line 328: The average (ozone) concentration

R: Thanks for your correction. We have modified it, please refer to Line 357.

Comments:

(1) For the title: "Why Did Ozone Concentrations Increase During Shanghai's Static Management?" does it precise to say increase? As the fact is that the $O_3$ concentration keeps in high level during the static period, instead of an increasement.

R: Thanks for your comment. We agree with your statement, it is a rigorous statement. We have changed the title to "Why Did Ozone Concentrations Remain High During Shanghai's Static Management? A Statistical and Radical Chemistry Perspective".

(2) In line 136-138: "The literatures have shown that Shanghai in the spring largely operates under VOCs-limited regime (Li et 137 al., 2021a; Xue et al., 2022). Therefore, the reduction in VOCs during the static management period may not be enough to counteract the titration effect of NOx, and may even alter the ozone formation regime in Shanghai." What is the final conclusion for this prediction based on the current work, does the ozone formation regime changed in the static period?

R: Thanks for your comment. Based on the existing analysis, the free radical chemistry simulation results indicate that such a proportional decrease in precursors in Shanghai would enhance the radical cycling without altering the ozone formation regime.

(3) In line 179-181, four type clusters were described according to the background concentration and net production of ozone, can the authors provide more information on how these two parameters were obtained?

R: Thanks for your comment. We define the minimum concentration in the ozone diurnal profile as the background concentration, and the difference between the midday peak and the morning trough represents the net ozone production. By examining the ozone diurnal profiles of the four clusters, we can determine these two parameters. In this study, the background concentration and net generation of Cluster 1 are approximately 23 ppbv and 15 ppbv, respectively; for Cluster 2, they are 17 ppbv and 46 ppbv; for Cluster 3, 41 ppbv and 17 ppbv; and for Cluster 4, 33 ppbv and 51 ppbv, respectively.

We have included the information about these two parameters in the manuscript, please refer to Line 190-196.

(4) In line 192-193, it says: "Purely statistical analysis indicated that the significant increase in ozone levels in 2022 was due to a higher proportion of Cluster 3 and Cluster 4, which had higher ozone concentrations during the static management period.", Maybe it is not good to say "significant increase in ozone level", maybe "significant high and stable level of ozone" better?

R: Thank you for your constructive feedback. We appreciate your suggestions and have made the necessary revisions. Please refer to Line 206-207.

(5) In line 165-166, it says: "Consequently, the rise in OVOC proportion during the static management period has the potential to enhance the photochemical process". However, when the simulation results presented in line 237-267, the influence of the proportion change of OVOCs in 2022 on $O_3$ was not mentioned, can the conclusion be draw according to the model results?

R: Thanks for your comment. VOCs are both a source of radicals and a consumable in the propagation of radicals. The effect of changes in the concentrations and ratios of the VOCs components on the photochemical processes was analyzed from the point of view of the sources and sinks of the radicals. Figure R2 shows the proportion and daily variation of primary sources of radicals. We can see that in 2020, 2021 and 2023, the primary sources involved in the photolysis process were the main ones, accounting for 81.7%. 71.9% and 83.5%, and reactions with VOCs (mainly alkenes) accounted for 18.3%, 28.1% and 26.5% respectively. The results show that although the decrease in VOCs concentration in 2022 has led to a decrease in the intensity of reactions with VOCs as primary sources of radicals, the proportion of reactions with VOCs was close to the level in 2021. This shows that the reaction with VOCs to generate radicals was still important during the static management period. In addition, the proportions of OVOCs photolysis in 2020, 2021 and 2022 were 14.3%, 19.6% and 10.6% respectively. The photolysis of OVOCs in 2022 was lower in level and proportion than in 2020 and 2021, so from the perspective of radical sources, there is no direct evidence that the change in the concentration proportion of OVOCs during static management would enhance the photochemical process.

[Figure]

**Figure R2. The proportions (a, b, c) and the mean diurnal profiles (d, e, f) of primary sources of daytime radicals during the periods from April to May of 2020, 2021, and 2022.**

The radical cycle begins with the degradation of VOCs triggered by OH radicals. A widely used

indicator of atmospheric oxidative capacity is the OH reactivity ($k$OH), defined as the reaction rate coefficient multiplied by the concentration of OH reactants, depending on the abundance and composition of major pollutants. Figure R3 illustrates the proportions of OH reactivity contributed by different VOCs components and its daily variation. In 2020, 2021 and 2022, the total $k$OH contributed by VOCs was 2.1 s$^{-1}$, 2.6 s$^{-1}$ and 1.1 s$^{-1}$, respectively. Due to the influence of static management, the different proportions of decline of different components led to changes in the contribution to $k$OH. The contribution of aromatics decreased from 27% in 2020 and 31.6% in 2021 to 12% in 2022, while the contribution of olefins increased from 40.8% in 2020 and 43.3% in 2021 to 54.2% in 2022. And the contribution of OVOC did not change significantly. Alkenes play a large role from a radical propagation perspective, but there is no direct evidence that changes in OVOC concentration proportions would enhance photochemical processes.

[Figure]

**Figure R3. The proportions (a, b, c) and the mean diurnal profiles (d, e, f) of kOH for different VOCs components during the periods from April to May of 2020, 2021, and 2022.**

Therefore, it is rather arbitrary to judge in the manuscript that an increase in the proportion of OVOC concentration is likely to enhance photochemical processes just from the change in concentration. Thanks for your comments, we have revised our hypothesis about the role of OVOCs in the manuscript. Please refer to Line 174-176.

"*In the radical chemistry section, the photolysis of OVOCs, as well as the reactions of O₃ and NO₃ with VOCs, have been quantified for their contributions to the radicals. Additionally, the role of VOCs in the propagation of radicals has been quantified.*".

(6) The information in line 256-257 is for the total P(ROx), but it was not shown in the Figure 6, maybe it should be added to directly see? The same for line 279-280, the total daily P(ROx) was not shown in Figure d-e for different clusters.

R: Thanks for your comment. Following your suggestion, Thank you for your comments. In accordance with your suggestions, we have replaced Figures 6 and 7 in the manuscript with Figures R4 and R5, respectively, to include the total primary sources. Please refer to Line 290 and Line 316. Furthermore, we have included Figures R6 and R7 in the Supplement. Please refer to the Figure S13 and Figure S14. Those include the proportion of each major source and the average

daily profile of the total daily P(ROx).

Moreover, we have included an explanation in the Lines 277-278 stating, "*Overall, the total primary production rates of ROx in 2022 were 2.34 ppbv h$^{-1}$, which is lower than the 2.94 ppbv h$^{-1}$ and 2.85 ppbv h$^{-1}$ in 2021, as depicted in Figure S13 of the Supplement.*".

[Figure]

**Figure R4. The mean diurnal profiles of simulated OH (a), HO$_2$ (e), and RO$_2$ (i) concentrations in 2020, 2021, and 2022. Colored areas denote 95% confidence intervals; The mean diurnal profiles of primary sources of OH radical (b-d), HO$_2$ radical (f-h), and RO$_2$ radical (j-l) from model calculations in 2020, 2021 and 2022.**

[Figure]

**Figure R5. The mean diurnal profiles of simulated OH (a), HO$_2$ (b), and RO$_2$ (c) concentrations for Cluster 1, Cluster 2, Cluster 3, and Cluster 4. Colored areas denote 95% confidence intervals; The mean diurnal profiles of primary sources of ROx radical (d-g) from model calculations for Cluster 1, Cluster 2, Cluster 3, and Cluster 4.**

[Figure]

**Figure R6. The proportions (a, b, c) and the mean diurnal profiles (d, e, f) of primary sources of daytime radicals during the periods from April to May of 2020, 2021, and 2022.**

[Figure]

**Figure R7. The proportions (a, b, c, d) and the mean diurnal profiles (e, f, g, h) of primary sources of daytime radicals for Cluster 1, Cluster 2, Cluster 3, and Cluster 4.**

(7) According to the current analysis, it seems like that the radical chemistry analysis based on the clusters do not really help to explain the high $O_3$ concentration in 2022, because in line 278-279, it says that:"Cluster 2 and Cluster 4, characterized by significant net ozone production, exhibit distinct features in radical chemistry", however, in 2022, the main clusters are 3 & 4, in addition, in figure 7, the mean diurnal profiles of OH and $HO_2$ for cluster 3 are always above cluster 4, and even for $RO_2$, it seems like the average concentrations for $RO_2$ in cluster 3 and 4 are close by eyes, please check again if there is a big difference of ROx or P(ROx) for cluster 3 and 4.

R: Thanks for your comment. We have characterized the ozone profiles after clustering into four distinct clusters: Cluster 1 with low background concentration and low net production; Cluster 2 with low background concentration and high net production; Cluster 3 with high background concentration and low net production; and Cluster 4 with high background concentration and high net production, as shown in Table R1. Upon statistical analysis, it was found that the number of days with higher average ozone concentrations in Cluster 3 and Cluster 4 increased significantly in 2022 compared to 2020 and 2021.

As shown in Table R1, the reasons for the high average ozone concentration levels in these two clusters were not entirely the same. The high concentrations in Cluster 3 were due to elevated

ozone background concentration, whereas in Cluster 4, the high concentrations were attributed to both high background values and high net production. For Cluster 2, it had a high net ozone production but a lower background concentration. Therefore, in the analysis of the primary sources of radicals, Cluster 2 and Cluster 4, which had high net production, exhibited significantly higher primary sources of radicals compared to Cluster 1 and Cluster 3. This suggests from the radical chemistry analysis based on the clusters that a high net production indicates active photochemical processes.

The cluster-based analysis of radical chemistry can explain the high $O_3$ concentrations observed in 2022. Cluster 4 is characterized by higher concentrations of $HO_2$ and $RO_2$ radicals and a higher primary source of radicals. Cluster 3 has moderate levels of radical concentrations and primary sources of radicals, but it features a high ozone background value. The increase in the number of days associated with these two clusters in 2022 has contributed to the elevated ozone levels. Cluster 2 has a higher primary source of radicals, indicating active photochemical processes, but the overall ozone levels are moderate due to the lower background ozone values.

**Table R1. The ozone concentration (ppbv) characteristics, the average daytime radicals concentration (molecules cm$^{-3}$), and the average reaction rates of the main processes for Cluster 1-Cluster 4.**

|  | Cluster 1 | Cluster 2 | Cluster 3 | Cluster 4 |
|---|---|---|---|---|
| $O_3$ background | 22.4 | 16.9 | 40.6 | 33.3 |
| $O_3$ net production | 14.8 | 45.9 | 17.5 | 50.9 |
| OH concentration | $1.4\times10^6$ | $3.3\times10^6$ | $3.1\times10^6$ | $3.9\times10^6$ |
| $HO_2$ concentration | $0.6\times10^8$ | $1.5\times10^8$ | $1.9\times10^8$ | $3.5\times10^8$ |
| $RO_2$ concentration | $0.3\times10^8$ | $0.8\times10^8$ | $0.9\times10^8$ | $2.2\times10^8$ |
| P(ROx) | 0.92 | 2.09 | 1.35 | 2.23 |
| OH propagation (OH+VOCs) | 0.68 | 1.75 | 1.16 | 2.03 |
| OH termination (OH+NO$_2$) | 0.63 | 1.14 | 0.69 | 0.88 |
| propagation/termination | 1.08 | 1.54 | 1.69 | 2.29 |
| P($O_3$) | 3.12 | 7.73 | 5.24 | 8.20 |

(8) In line 300: "OH oxidation of CO and VOCs produces $HO_2$ and $RO_2$", the concentration of CO was not given. And also, the concentration of NO was not given.

R: Thanks for your comment. The CO and NO data you referred to were not observed simultaneously in this study. Consequently, the CO data used in the model was a fixed value of 0.8 ppmv, and the NO data was calculated by the OBM model. This indeed represents a source of uncertainty in the model.

(9) In line 306-308: Specifically, the reactions of OH+NO$_2$ and RO$_2$+NO$_2$ accounted for approximately 0.51 ppbv h$^{-1}$ (0.76 ppbv h$^{-1}$ in 2020 and 0.74 ppbv h$^{-1}$ in 2021) and 0.80 ppbv h$^{-1}$ (0.96 ppbv h$^{-1}$ in 2020 and 1.55 ppbv h$^{-1}$ in 2021) of the ROx radical loss on daytime average, respectively." However, the reaction of RO$_2$ with NO$_2$ is generally considered to be not important as the product RO$_2$NO$_2$ formed in the reaction of RO$_2$+NO$_2$ for R=alkyl or substituted alkyl (but not for R=acyl) will thermally decompose rapidly back to reactants at around room temperature

(Atkinson, R., Atmospheric chemistry of VOCs and NOx. Atmospheric Environment, 2000. 34(12): p. 2063-2101.), does the decomposition reaction of $RO_2NO_2$ included in the model? This process was not shown in Figure 8.

R: Thanks for your comment. Firstly, the mcm mechanism utilized in the model includes the decomposition reactions of $RO_2NO_2$. Table R2 displays a selection of these $RO_2NO_2$ decomposition reactions along with their respective rate constants. Secondly, the manuscript initially described only the reaction rate between $RO_2$ and $NO_2$, which could be misleading. Consequently, we have re-evaluated the net production rate of $RO_2NO_2$, calculated as the generation rate minus the decomposition rate. The findings are now displayed in the revised ==Figure 8.==

**Table R2. $RO_2NO_2$ decomposition reactions involved in MCM mechanism.**

| | |
|---|---|
| reaction rate constant | KD0 = 1.10D-05*M*EXP(-10100/TEMP) ;
KDI = 1.90D17*EXP(-14100/TEMP) ;
KRD = KD0/KDI ;
FCD = 0.30 ;
NCD = 0.75-1.27*(LOG10(FCD)) ;
FD = 10@(LOG10(FCD)/(1+(LOG10(KRD)/NCD)**2)) ;
KBPAN = (KD0*KDI)*FD/(KD0+KDI) ; |
| reaction equations | % KBPAN : ACRPAN = $ACO_3$ + $NO_2$ ;
% KBPAN : PAN = $CH3CO_3$ + $NO_2$ ;
% KBPAN : BZEMUCPAN = $BZEMUCCO_3$ + $NO_2$ ;
% KBPAN : TLEMUCPAN = $TLEMUCCO_3$ + $NO_2$ ;
% KBPAN : OXYMUCPAN = $OXYMUCCO_3$ + $NO_2$ ;
% KBPAN : MXYMUCPAN = $MXYMUCCO_3$ + $NO_2$ ;
% KBPAN : PXYMUCPAN = $PXYMUCCO_3$ + $NO_2$ ;
% KBPAN : EBZMUCPAN = $EBZMUCCO_3$ + $NO_2$ ;
% KBPAN : PBZMUCPAN = $PBZMUCCO_3$ + $NO_2$ ;
% KBPAN : IPBZMUCPAN = $IPBZMUCCO_3$ + $NO_2$ ;
… |

(10) In addition to the $OH-RO_2-HO_2$ cycle, which leads to a high production rate of ROx, and the other cycle for $OH-HO_2$ seems like also be further strengthened in 2022. The average rates for each step related to $OH-HO_2$ cycle are summarized in the following table according to the model results, it is obvious that reactions for $HO_2$ + R (R=NO, $O_3$ and $NO_3$) → OH and OH + R (R=CO, HCHO) → $HO_2$ in 2022 with higher rates. Especially for reaction related to $O_3$, which is 2.6 and 2.1 times faster in 2022 than in 2020 and 2021. Will the $OH-HO_2$ cycle also help to explain the high ROx or P(ROx) in 2022?

| | $HO_2$+R→OH | | | $HO_2$←OH+R | |
|---|---|---|---|---|---|
| R | NO | $O_3$ | $NO_3$ | CO | HCHO |
| 2020 | 2.761 | 0.047 | 0.013 | 1.442 | 0.225 |
| 2021 | 3.215 | 0.059 | 0.013 | 1.449 | 0.185 |
| 2022 | 3.352 | 0.122 | 0.018 | 2.108 | 0.275 |
| 2022/2020 | 1.2 | 2.4 | 1.4 | 1.1 | 1.2 |
| 2022/2021 | 1.0 | 2.0 | 1.3 | 1.2 | 1.5 |

R: Thanks for your comment. The insights you've raised are indeed constructive. As previously mentioned in the text, the report by Volkamer et al. (2010) suggests that approximately 20% of radical production is attributed to the decomposition of closed-shell species, while 80% originates from radical cycles. Upon discovering the improper handling of photolysis data, we reran the model. The table below compiles the average rates for each step related to the OH-HO$_2$ cycle based on the model's results. The results indicate that in 2022, the reaction rate for OH + R (R = CO, HCHO) → HO$_2$ was higher, while the rate for HO$_2$ + R (R = NO, O$_3$, and NO$_3$) → OH was lower compared to the years 2020 and 2021. As for reactions related to O$_3$, the rates in 2022 were 2.4 times and 2.0 times faster than in 2020 and 2021, respectively. We did not observe a more efficient OH-HO$_2$ cycle in 2022. Moreover, the total P(ROx) in 2022 was the lowest among the three years, yet the ROx concentration was the highest. The relationship between the generation rate and the concentration of products is not a direct cause-and-effect.

We have taken into account the OH-HO$_2$ cycle in the radical chemistry analysis, and the results of this analysis have been included in the manuscript. Please refer to Lines 332-334.

**Table R2. The reaction rates of the main processes in the OH-HO$_2$ cycle for the three years.**

| R | HO$_2$+R→OH | | | HO$_2$←OH+R | |
|---|---|---|---|---|---|
|  | NO | O$_3$ | NO$_3$ | CO | HCHO |
| 2020 | 3.911 | 0.052 | 0.014 | 1.904 | 0.309 |
| 2021 | 4.074 | 0.062 | 0.015 | 1.773 | 0.221 |
| 2022 | 3.435 | 0.125 | 0.019 | 2.086 | 0.274 |
| 2022/2020 | 0.9 | 2.6 | 1.4 | 1.1 | 0.9 |
| 2022/2021 | 0.8 | 2.1 | 1.3 | 1.2 | 1.2 |

**Reference:**

Volkamer, R., Sheehy, P., Molina, L. T., and Molina, M. J.: Oxidative capacity of the Mexico City atmosphere-Part 1: A radical source perspective, Atmospheric Chemistry and Physics, 10, 6969-6991, https://doi.org/10.5194/acp-10-6969-2010, 2010.